# Peer Prediction for Learning Agents

**Shi Feng**
Institute for Interdisciplinary Information Sciences, Tsinghua University
Beijing, China
fengs19@mails.tsinghua.edu.cn

**Fang-Yi Yu**
Department of Computer Science, George Mason University
Fairfax, VA, USA
fangyiyu@gmu.edu

**Yiling Chen**
John A. Paulson School of Engineering and Applied Sciences, Harvard University
Cambridge, MA, USA
yiling@seas.harvard.edu

## Abstract

Peer prediction refers to a collection of mechanisms for eliciting information from human agents when direct verification of the obtained information is unavailable. They are designed to have a game-theoretic equilibrium where everyone reveals their private information truthfully. This result holds under the assumption that agents are Bayesian and they each adopt a fixed strategy across all tasks. Human agents however are observed in many domains to exhibit learning behavior in sequential settings. In this paper, we explore the dynamics of sequential peer prediction mechanisms when participants are learning agents. We first show that the notion of no regret alone for the agents' learning algorithms cannot guarantee convergence to the truthful strategy. We then focus on a family of learning algorithms where strategy updates only depend on agents' cumulative rewards and prove that agents' strategies in the popular Correlated Agreement (CA) mechanism converge to truthful reporting when they use algorithms from this family. This family of algorithms is not necessarily no-regret, but includes several familiar no-regret learning algorithms (e.g multiplicative weight update and Follow the Perturbed Leader) as special cases. Simulation of several algorithms in this family as well as the $\epsilon$-greedy algorithm, which is outside of this family, shows convergence to the truthful strategy in the CA mechanism.

## 1 Introduction

A fundamental challenge in many domains is to elicit high-quality information from people when directly verifying the acquired information is *not* feasible, either because the ground truth is not available or because it is too costly to obtain. Notable settings include asking people to label data for machine learning, having students perform peer grading in education, and soliciting customer feedback for products and services.

The peer prediction literature has made impressive progress on this challenge in the past two decades, with many mechanisms that have desirable incentive properties developed for this problem [20, 19, 27, 5, 22, 8, 14, 25, 17, 16, 21, 18, 23, 15]. The term peer prediction refers to a collection of reward

36th Conference on Neural Information Processing Systems (NeurIPS 2022).

mechanisms that solicit information from human agents and reward each agent solely based on how the agent's reported information compares with that of the other agents, without having access to the ground truth. Under some assumptions, many peer prediction mechanisms [20, 19, 27, 22, 8, 16, 21] guarantee that every agent truthfully reporting their information is a game-theoretic equilibrium, and the more recent multi-task peer prediction mechanisms [5, 14, 25, 17, 18, 23, 15] further ensure that agents receive the highest expected payoff at the truthful equilibrium, compared with other strategy profiles.

While achieving truthful reporting as a highest-payoff equilibrium is a victory to declare for this challenging without-verification setting, there are however caveats associated with adopting the notion of equilibrium as a solution concept. The equilibrium results rely on the assumption that participants are fully rational Bayesian agents. Equilibrium is a static notion and does not address how agents, who act independently, jump to play their equilibrium strategies. Moreover, the equilibrium results of multi-task peer prediction mechanisms heavily depend on a *consistent strategy* assumption, that is, each agent is assumed to adopt a fixed strategy across all tasks in which she participates. All together these assumptions exclude the possibility that agents may explore and learn from previous experience, a behavior that's not only commonly observed in practice but also has been modeled in studying other strategic settings [3, 6, 4].

This paper is the first theoretical study on the dynamics of sequential peer prediction mechanisms when participants are learning agents. The main question that we explore is whether and when in sequential peer prediction, learning agents will converge to all playing the truthful reporting strategy. We first consider agents adopting no-regret learning algorithms and prove that the notion of no regret alone cannot guarantee convergence to truthful reporting. We then define a natural family of reward-based learning algorithms where strategy updates only depend on agents' cumulative rewards. While algorithms in this family are not necessarily no-regret (e.g. the Follow the Leader algorithm), this family includes some familiar no-regret learning algorithms, including the Multiplicative Weight Update and the Follow the Perturbed Leader algorithms. Our main result shows that, for the binary-signal setting, agents' strategies in the popular Correlated Agreement (CA) mechanism [5] converge to truthful reporting when agents use any algorithm from this family. To prove the result, we show the process has a self-fulfilling property: once Alice and Bob have large accumulated rewards for truth-telling, they are more likely to play truth-telling and resulting in larger accumulated rewards. Theoretically, we carefully partition the process into three stages, bad, intermediate, and good events illustrated in fig. 1, and use tools in martingale theory to argue the progress of the process. Finally, we simulate the strategy dynamics in the CA mechanism for several algorithms in this family as well as for the $\epsilon$-greedy algorithm, which doesn't belong to this family. We observe convergence to truthful reporting for all algorithms considered in our simulation, suggesting an interesting future direction to characterize all learning algorithms that converge to truthful reporting.

**Related Works**  This paper relates to two lines of work, information elicitation and mechanisms for learning agents.

**Information Elicitation Mechanisms** The literature on information elicitation without verification focuses on capturing the strategic aspect of human agents. In multi-task settings, Dasgupta and Ghosh [5] proposed a seminal informed truthful mechanism, the Correlated Agreement (CA) mechanism, for binary positively correlated signals. A series of works then relaxed the binary and positively correlation assumptions [25, 17, 23, 15]. Additionally, Zheng et al. [28] study the limitation of information elicitation in the multi-task setting. However, all of the above works assume agents using consistent strategies that are identical across all tasks. The consistent strategy assumption excludes the possibility of agent learning. Our work removes the consistent strategy assumption and explicitly considers learning agents. We theoretically prove truthful convergence of the CA mechanism when agents using algorithms from a family of reward-based online learning algorithms.

Our work of considering learning agents can be viewed as a way of testing the robustness of information elicitation mechanisms with respect to deviation from the rational Bayesian agent model. From this perspective, Shnayder et al. [26] is closely related to ours. They consider sequential information elicitation and empirically study if agents using replicator dynamics can converge to truth-telling in the CA mechanism and several other mechanisms. Our work theoretically proves that, besides replicator dynamics, learning agents can converge to truth-telling in the CA mechanism when they use a general family of learning algorithms. Additionally, Schoenebeck et al. [24] designed an information elicitation mechanism that was robust against a small fraction of adversarial agents.

**Mechanisms for Learning Agents** Several works in economics and computer science try to design mechanisms for learning agents, rather than for rational, Bayesian ones. Braverman et al. [3] studied pricing mechanisms for learning agents with no external regrets. Their work was generalized by Deng et al. [6] to consider repeated Stackelberg games in full-information settings. Camara et al. [4] further proposed counterfactual internal regrets (CIR) together with no-CIR assumption, which was proved to be a sufficient behavior assumption for no-regret principal mechanism design in repeated stage games. However, all of these works focus on a single agent or full-information games, while peer prediction is an incomplete-information game with multiple agents. Finally, our goal is slightly different from that of most sequential mechanism design. Instead of maximizing the mechanism designer's utility, our goal is to incentivize truthful reporting from agents.

## 2  Peer Prediction Settings

For simplicity we consider two agents[1], Alice and Bob, who work on a sequence of tasks indexed by $t \geq 1$. For round $t$, both agents work on task $t$, Alice receives a signal $X_t = x_t$ in $\{0, 1\}$, and Bob a signal $Y_t = y_t$ in $\{0, 1\}$ where $X_t$ and $Y_t$ denote random variables, and $x_t$ and $y_t$ are their realizations. Then Alice and Bob report $\hat{X}_t = \hat{x}_t$ and $\hat{Y}_t = \hat{y}_t$ in $\{0, 1\}$. We define $\mathbf{X}_{\leq t} = \{X_s : 1 \leq s \leq t\} \in \{0, 1\}^t$ and $\hat{\mathbf{X}}_{\leq t} = \{\hat{X}_s : 1 \leq s \leq t\} \in \{0, 1\}^t$ to denote Alice's signal profile and report profiles until $t$-th round respectively, and define $\mathbf{Y}_{\leq t}$ and $\hat{\mathbf{Y}}_{\leq t}$ for Bob similarly. We use $\mathbf{X}, \hat{\mathbf{X}}, \mathbf{Y}$, and $\hat{\mathbf{Y}}$ for the complete signal and report profiles. Additionally, we consider the signals are generated from some distribution $\mathbb{P}$ that satisfies the following assumptions:

**Assumption 2.1** (A priori similar tasks [5]). Each pair of signal is identically and independently (i.i.d.) generated: there exists a distribution $P_{X,Y}$ over $\{0, 1\}^2$ such that $(X_t, Y_t) \sim P_{X,Y}$ for any $t \in \mathbb{N}^+$. Moreover, we assume the distribution has full support, $P_{X,Y}(x, y) > 0$ for all $x, y \in \{0, 1\}$.

**Assumption 2.2** (Positively correlated signals). The distribution $P_{X,Y}$ is positively correlated, $\min\{P_{X,Y}(1, 1), P_{X,Y}(0, 0)\} > \max\{P_{X,Y}(1, 0), P_{X,Y}(0, 1)\}$.

Now we introduce multi-task peer prediction mechanisms and sequential peer prediction mechanisms, and their relation. We will focus on the sequential setting. Multi-task peer prediction mechanisms work on a fixed number of tasks. Formally, a multi-task peer prediction mechanism on $k$ tasks is a pair of payment functions $\bar{M} : \{0, 1\}^k \rightarrow [0, 1]^2$. For instance, the *(multi-task) correlated agreement mechanism* (CA mechanism)[2] [5, 25, 26] is $\bar{M}^{CA}(\hat{\mathbf{x}}, \hat{\mathbf{y}}) = (\mathbb{I}[\hat{x}_2 = \hat{y}_2] - \mathbb{I}[\hat{x}_2 = \hat{y}_1], \mathbb{I}[\hat{y}_2 = \hat{x}_2] - \mathbb{I}[\hat{y}_2 = \hat{x}_1])$ for all $\hat{\mathbf{x}}, \hat{\mathbf{y}} \in \{0, 1\}^2$. Intuitively, the CA mechanism rewards agreement on the same task and punishes agreement on uncorrelated tasks.

A *sequential information elicitation mechanism* is a sequence of payment functions $\mathcal{M} = \{M_t : t \geq 1\}$ where $M_t : \{0, 1\}^{2 \times t} \rightarrow [-1, 1]$ for all $t$. After Alice and Bob reporting $\hat{\mathbf{x}}_{\leq t}$ and $\hat{\mathbf{y}}_{\leq t}$ in round $t$, the mechanism computes $(r_t, s_t) := M_t(\hat{\mathbf{x}}_{\leq t}, \hat{\mathbf{y}}_{\leq t})$ and pay $r_t$ to Alice and $s_t$ to Bob. Here we assume $M_t$ can only depends on a constant $k$ round of reports so that $M_t(\hat{\mathbf{x}}_{\leq t}, \hat{\mathbf{y}}_{\leq t}) = M_t(\hat{x}_{t-k+1}, \hat{x}_{t-k+1}, \ldots, \hat{x}_t, \hat{y}_{t-k+1}, \hat{y}_{t-k+1}, \ldots, \hat{y}_t)$ for all $t$, $\hat{\mathbf{x}}_{\leq t}$, and $\hat{\mathbf{y}}_{\leq t}$, and we call such $\mathcal{M}$ *rank $k$ mechanism*. For instance, the (multi-task) CA mechanism can be adopted as a sequential rank 2 information elicitation mechanism: At round $t$, the payment is

$$M_t^{CA}(\hat{\mathbf{x}}_{\leq t}, \hat{\mathbf{y}}_{\leq t}) = (\mathbb{I}[\hat{x}_t = \hat{y}_t] - \mathbb{I}[\hat{x}_t = \hat{y}_{t-1}], \mathbb{I}[\hat{y}_t = \hat{x}_t] - \mathbb{I}[\hat{y}_t = \hat{x}_{t-1}]) \tag{1}$$

where $\hat{x}_0$ and $\hat{y}_0$ are set as 0. Similarly, we say a sequential information elicitation mechanism $\mathcal{M} = (M_t)_{t \geq 1}$ is a *sequential version* of a multi-task information elicitation mechanism $\bar{M}$ if $M_t(\hat{\mathbf{x}}_{\leq t}, \hat{\mathbf{y}}_{\leq t}) = \bar{M}(\hat{x}_{t-k+1}, \hat{x}_{t-k+1}, \ldots, \hat{x}_t, \hat{y}_{t-k+1}, \hat{y}_{t-k+1}, \ldots, \hat{y}_t)$ for all $\hat{\mathbf{x}}_{\leq t}, \hat{\mathbf{y}}_{\leq t}$ and $t \geq k$. The payment at round $t \geq k$ is $\bar{M}$ on the latest $k$ reports. Conversely, a sequential information elicitation mechanism can be seen as a sequence of multi-task information elicitation mechanisms.

---

[1]For more than two agents, we can partition the agents into groups of two agents to run our mechanisms when the number of agents is even. Then all our results still hold. Finally, when the number of agents is odd, we can pair the unpaired agent with a reference agent whose payment is not affected by the unpaired one.

[2]While the CA mechanism can be defined on non binary setting and does not require positive correlation. [25], with assumption 2.2, the CA mechanism reduces to eq. (1) and is first proposed in [5]. Finally, note that when the number of task is greater than two, we can compute the payment based on the last two tasks or two random tasks since agents using consistent strategy and assumption 2.1.

Now we formally define agents' strategies. Due to symmetry, we introduce notation for Alice and omit Bob's. Given an information elicitation mechanism $\mathcal{M}$, at round $t$, Alice observes her signal $x_t$ and decides on her report $\hat{x}_t$. Thus, Alice has four *options (pure strategies)*: 1) $\text{opt}_1$: report the private signal truthfully, 2) $\text{opt}_2$: flip the private signal, 3) $\text{opt}_3$: report 1 regardless of the signal, and 4) $\text{opt}_4$: report 0 regardless of the signal. We call $\text{opt}_3$ and $\text{opt}_4$ uninformative strategies. We use $\text{opt}_t^X$ to denote Alice's pure strategy, and $r_t$ for her payoff at round $t$. At each round $t$, Alice knows her previous signals $\mathbf{x}_{\leq t} \in \{0, 1\}^t$, her pure strategies $\text{opt}_1^X, \ldots, \text{opt}_{t-1}^X$, and Bob's reports $\hat{\mathbf{y}}_{\leq t-1}$, so we use $\mathcal{F}_t = \{\mathbf{x}_{\leq t}, \text{opt}_{\leq t-1}^X, \hat{\mathbf{y}}_{\leq t-1}\}$ to denote Alice knowledge at round $t$. Thus, Alice's mixed strategy at round $t$ is a stochastic mapping $\sigma_t^X$ from $\mathcal{F}_t$ to $\{\text{opt}_1, \text{opt}_2, \text{opt}_3, \text{opt}_4\}$. We'll abuse our nation and also use $\sigma_t^X = \text{opt}_t^X$ to represent the realized pure strategy. Finally, a learning algorithm of Alice is a mapping from an information elicitation mechanism $\mathcal{M}$ to her strategies.

**Strong Truthfulness for Rational and Bayesian agents**    Previous works on information elicitation try to ensure truth-telling $\text{opt}_1$ is the best strategy for rational and strategic agents. In particular, a mechanism is *strongly truthful* if Alice and Bob report truthfully is a Bayesian Nash Equilibrium (BNE) and they get strictly higher payment at this BNE than at any other non-permutation BNE. We present the formal definitions in the appendix. Informally, in a permutation BNE, every agent's strategy on each round is a permutation/bijection from his/her signals to reports. However, the equilibrium results of previous mechanisms not only require assumption 2.1 but further assume agents using *consistent strategies*. Specifically, Alice uses a consistent strategy if there is a fixed distribution on $\{\text{opt}_1, \text{opt}_2, \text{opt}_3, \text{opt}_4\}$ so that $\text{opt}_t^X$ is generated from a fixed distribution that is independent of her private signals on other tasks and the round number. For instance, when Alice and Bob are Bayesian and use consistent strategies under assumption 2.2 and 2.1, Dasgupta and Ghosh [5] show CA mechanism in eq. (1) is strongly truthful. Intuitively, positive correlation assumption 2.2 guarantees that truthful reporting can maximize the chance of agreeing with the peer on the same task while avoiding agreeing on reports on other tasks. Furthermore, in appendix B we show CA mechanism merely has three types Bayesian Nash equilibria, at which both agents 1) play truth-telling $\text{opt}_1$, 2) flip the signal $\text{opt}_2$, or 3) generate uninformative reports (mixture between $\text{opt}_3, \text{opt}_4$) when agents use consistent strategies and assumptions 2.1 and 2.2 hold.

**Truthful Convergence for Learning Agents**    However, as we consider agents using a family of online learning algorithms to decide their strategies, standard solution concepts like Bayesian Nash equilibrium no longer apply. Additionally, online learning algorithms often have exploration, so we cannot hope agents will always use the truth-telling strategy. For learning agents, our goal is to test whether existing mechanisms can ensure that agents will *converge* to truthful reporting when they deploy certain learning behavior that goes beyond obliviously consistent strategies.

We now formalize the convergence of algorithms to truthful reporting. Because we want to elicit information without verification, it is information-theoretically impossible for us to separate permutation equilibrium, where all agents play $\text{opt}_2$, from truthful equilibrium, where all agents play $\text{opt}_1$, without any additional information [17]. However, if we have an additional bit of information on whether the prior of $0$ is larger than $1$, we may tell apart these two equilibria. We hence define convergence to truthful reporting as the limits of both $\text{opt}_t^X$ and $\text{opt}_t^Y$ being truth-telling ($\text{opt}_1$) or flipping ($\text{opt}_2$). Note that definition 2.3 requires almost surely convergence which is very strong convergence concept.

**Definition 2.3.** An information elicitation mechanism $\mathcal{M}$ achieves *truthful convergence* for agents using algorithms $A_1$ and $A_2$ respectively if and only if both sequences of pure strategies converge to truth-telling or both flipping the reports.

$$\Pr\left\{ \lim_{t \to +\infty} \text{opt}_t^X = \lim_{t \to +\infty} \text{opt}_t^Y = \text{opt}_1 \lor \lim_{t \to +\infty} \text{opt}_t^X = \lim_{t \to +\infty} \text{opt}_t^Y = \text{opt}_2 \right\} = 1.$$

## 3   Online Learning Algorithms

In this section, we explore candidates to model agents' learning behavior to replace Bayesian agents' consistent strategies in the literature. We first show the conventional no-regret assumption is a necessary but not a sufficient condition for truthful convergence in section 3.1. Then in section 3.2, we introduce a family of reward-based online learning algorithm to model agents' learning behavior,

and show that the family of reward-based online learning algorithms contains several common no-regret algorithms as special cases.

## 3.1 No-regret Online Learning Algorithms

We now investigate the relationship between no regret and truthful convergence. First, we show general no-regret algorithms may not ensure truthful convergence (theorem 3.1). However, we show the converse is almost true (theorem 3.2): If truthful convergence happens, the agents do not have regret when the sequential mechanism is a sequential version of a strongly truthful mechanism.

Given a sequential information elicitation mechanism $\mathcal{M} = (M_t^X, M_t^Y)_{t \geq 1}$, signals $\mathbf{x}, \mathbf{y}$, and reports $\hat{\mathbf{x}}, \hat{\mathbf{y}}$, we define $r_{i,t} = M_t^X(\text{opt}_i(x_1), \ldots, \text{opt}_i(x_t), \hat{\mathbf{y}}_{\leq t})$ be the payoff when Alice uses strategy $\text{opt}_i$ and Bob's choices are unchanged. Then Alice's *regret* is $Reg^X(T) = \max_i \sum_{t \leq T} r_{i,t} - \sum_{t \leq T} r_t$. Finally, we say that Alice's and Bob's online learning algorithms are *no regret* (on $\mathcal{M}$) if $\mathbb{E}[Reg^X(T)] = \mathbb{E}[Reg^Y(T)] = o(T)$ over the randomness of signals and the algorithms, and we say Alice and Bob are no regret for short.

One may hope that no regret as a behavior assumption for agents is sufficient for achieving desirable outcome in a mechanism. However, the following theorem shows that we cannot have an information elicitation mechanism that achieves truthful convergence for all no-regret agents.

**Theorem 3.1.** *For any sequential information elicitation mechanism $\mathcal{M}$ of rank $k \in \mathbb{N}$, there exist no-regret algorithms for Alice and Bob so that $\mathcal{M}$ cannot achieve truthful convergence.*

The main idea of the proof is that the no-regret assumption cannot prevent Alice and Bob from colluding. In our counterexample, Alice and Bob decide on a no-regret sequence of reports regardless of their signals once the mechanism is announced. Technically, we use a probabilistic method to show the existence of a deterministic and no-regret sequence of strategies $(\text{opt}_t^X, \text{opt}_t^Y)_{t \geq 1}$ that consists of reporting 1 or 0 regardless of private signals, i.e. $\text{opt}_3$ or $\text{opt}_4$. The formal proof is in appendix C.1.

The notion of truthful convergence in definition 2.3 provides an ideal truthful guarantee to the mechanism designer. Here we show that truthful convergence also ensures no regret for agents when the sequential mechanism is a sequential version of a strongly truthful one-shot multi-task mechanism. That is, when the one-shot mechanism admits truthful reporting as a highest-payoff BNE.

For instance, if a pair of algorithms exhibits truthful convergence on the sequential CA mechanism (eq. (1)), they are also no regret (on the game). Intuitively, if Bob converges to the truth-telling $\lim_{t \to \infty} \text{opt}_t^Y = \text{opt}_1$, the average expected gain of Alice deviating to $\text{opt}_i$ is equal to the expected gain of deviating to $\text{opt}_i$ when Bob always tells the truth. The gain is non positive because the CA mechanism is strictly truthful by lemma B.4. Therefore, the expected regrets $\mathbb{E}[Reg^X(T)]$ and $\mathbb{E}[Reg^Y(T)]$ are small. Theorem 3.2 formalizes and extends the above idea to any strongly truthful multi-task information elicitation mechanism.

**Theorem 3.2.** *Let $\bar{M}$ be a strongly truthful multi-task information elicitation mechanism, and $\mathcal{M}$ be a sequential version of $\bar{M}$. If $\mathcal{M}$ achieve truthful convergence for Alice and Bob using algorithms $A_1$ and $A_2$ respectively, then Alice and Bob are no regret.*

## 3.2 Reward-based Online Learning Algorithms

As shown in theorem 3.1, the no-regret assumption allow does not guarantee truthful convergence. In this section, we introduce a general family of online learning algorithms, *reward-based online learning algorithm*, under the general full feedback bandit setting, and we will apply these algorithms in sequential information elicitation mechanisms later. Informally, a reward-based online learning algorithm decides each round's strategy using a fixed update function that depends only on the accumulative reward.

For simplicity, we only consider algorithms on four strategies $\{\text{opt}_1, \text{opt}_2, \text{opt}_3, \text{opt}_4\}$ which can be extended easily. Recall that the payoff of choosing option $\text{opt}_i, i \in [4]$ at round $t$ is $r_{i,j}$ when others' choices are unchanged. We denote the accumulated payoffs of these four options as $R_{i,t} = \sum_{j=1}^{t} r_{i,j}$ for $i \in [4]$. Symmetrically, we use $S_{i,t}$ and $s_{i,t}$ to represent accumulated payoffs and the payoff of turning to choose option $\text{opt}_i$ in the $t^{th}$ round for Bob. For example, for our specific peer

prediction game using CA mechanism in eq. (1), $r_{1,t} = \mathbb{I}[x_t = \hat{y}_t] - \mathbb{I}[x_t = \hat{y}_{t-1}]$ and therefore, $R_{1,t} = \sum_{j=1}^{t}(\mathbb{I}[x_t = \hat{y}_t] - \mathbb{I}[x_t = \hat{y}_{t-1}])$ for Alice.

We consider a family of *reward-based online learning algorithms* $\mathcal{A}$ that use an *update function* $f : \mathbb{R}^4 \to \triangle^3$, and choose $\text{opt}_i$ with probability $f_i(R_{1,t-1}, R_{2,t-1}, R_{3,t-1}, R_{4,t-1})$ for $i \in [4]$ in the $t^{th}$ round, where $f_i$ is the $i^{th}$ coordinate of $f$. A such mechanism based on accumulated payoffs is denoted by $A_f$. We have three assumptions for the update function $f$, which are all very natural. The first two require that $f$ is exchangeable and preserves ordering.

**Assumption 3.3** (Exchangeability of $f$). For any $R_1, R_2, R_3, R_4 \in \mathbb{R}$ and an arbitrary permutation of them $R_{i_1}, R_{i_2}, R_{i_3}, R_{i_4}$, $f_{i_j}(R_1, R_2, R_3, R_4) = f_j(R_{i_1}, R_{i_2}, R_{i_3}, R_{i_4})$ for all $j \in [4]$.

**Assumption 3.4** (Order preservation of $f$). For any $R_1, R_2, R_3, R_4 \in \mathbb{R}$ and suppose that $R_{i_1}, R_{i_2}, R_{i_3}, R_{i_4}$ is a non-increasing order of them, for $f$ we have $f_{i_1}(R_1, R_2, R_3, R_4) \geq f_{i_2}(R_1, R_2, R_3, R_4) \geq f_{i_3}(R_1, R_2, R_3, R_4) \geq f_{i_4}(R_1, R_2, R_3, R_4)$.

Finally we consider the strategy chosen by the update function $f$ when the accumulated payoff of an strategy is much higher than that of other strategies (assumption 3.5). Appendix D.2 shows that the assumption is necessary for reward-based online learning algorithms to achieve no regret for any online decision problem.

**Assumption 3.5** (Full exploitation of $f$). $\lim_{R_1 - \max\{R_2, R_3, R_4\} \to +\infty} f_1(R_1, R_2, R_3, R_4) = 1$.

Now we show that the family of reward-based online learning algorithms $\mathcal{A}$ satisfying assumptions 3.3 to 3.5 contains several classic no-regret online learning algorithms [10, 13].

**Theorem 3.6.** *$\mathcal{A}$ contains Follow the Perturbed Leader (FPL) algorithm, and Multiplicative Weights algorithm as special cases. Corresponding $f$'s for them are listed as below:*

**Multiplicative Weight algorithm** $f_i^{hedge\ 1}(R_1, R_2, R_3, R_4) = \frac{e^{\beta R_i}}{\sum_{j \in [4]} e^{\beta R_j}}$ *for $i \in [4]$*

**FPL algorithm** *Given a noise distribution $\mathcal{N}$ on scalars, $f_i^{FPL}(R_1, R_2, R_3, R_4) = \Pr\left\{R_i + p_i = \max_{j \in [4]}\{R_j + p_j\} \Big| p_j \overset{iid}{\sim} \mathcal{N}, j \in [4]\right\}$ for $i \in [4]$.*

In the binary peer prediction problem using CA mechanism, we can further show that replicator dynamics and linear updating multiplicative weight algorithm [2] are both in $\mathcal{A}$ in appendix E.

**Remark 3.7.** Our reward based online learning is very similar to the mean-based algorithm in [3] and both use the accumulated rewards to characterize the algorithm's choice. Additionally, like the mean-based algorithms, our reward based online algorithm may contain algorithms with regret in genral game, e.g., follow the leader. The family of reward based online algorithms uses an identical update function across all rounds. Thus some no-regret algorithms, e.g., $\epsilon$-greedy with time decreasing $\epsilon$, doesn't belong to family. Finally, if a mean-based algorithm is also a reward based algorithm that uses the same update function in each round, then its update function satisfies assumption 3.5.

## 4 Truthful Convergence of CA Mechanism on Reward-based Algorithms

Now we present our main result. We will show that the sequential CA mechanism can achieve truthful convergence if both agents use reward-based online learning algorithms from $\mathcal{A}$. This convergence result suggests that the classical CA is robust even when agents deviate from Bayesian rational behavior and use a general family of online learning algorithms.

**Theorem 4.1.** *Under assumptions 2.1 and 2.2, the binary-signal, sequential CA mechanism as defined in eq. (1) achieves truthful convergence when agents use reward-based algorithms $A_f$ and $A_g$, where the update functions $f$ and $g$ satisfy assumptions 3.3 to 3.5.*

Note that given agents' online learning algorithm and the payment function, the sequence of accumulated reward vector $(R_{1,t}, R_{2,t}, R_{3,t}, R_{4,t}, S_{1,t}, S_{2,t}, S_{3,t}, S_{4,t})_{t \geq 1}$ forms a stochastic process and we define $\mathcal{H}_t$ as the game history of the rewards and private signals in the first $t$ round. Additionally, if the accumulated reward of truth-telling $R_{1,t}, S_{1,t}$ is much larger than the others', we can show Alice and Bob converge to the truth-telling by assumption 3.5. Thus, it is sufficient for us to track the evolution of accumulated reward vector. Though the process of accumulated reward vector is not

a Markov chain because the payment function eq. (1) depends on reports in two rounds, we can still use ideas from semi-martingale to track the process.

Before proving our main result, we first present two properties of CA mechanism for our binary signal peer prediction problem. Lemma 4.2 shows that the accumulated payoffs of uninformative strategies $\text{opt}_3, \text{opt}_4$ $R_{3,t}, R_{4,t}, S_{3,t}$, and $S_{4,t}$ are always bounded.

**Lemma 4.2.** *Given the game defined in theorem 4.1, for any round $t$, the accumulated payoffs $R_{3,t}, R_{4,t}, S_{3,t}$, and $S_{4,t}$ for two agents are bounded by $[-1, 1]$.*

The second one, Lemma 4.3, tells us that the summation of accumulated payoffs of $\text{opt}_1$ and $\text{opt}_2$ is always fixed and is equal to the summation of accumulated payoffs of uninformative ones $\text{opt}_3$ and $\text{opt}_4$ for both agents. The proofs of these two lemmas are in appendix F.1.

**Lemma 4.3.** *Given the game defined in theorem 4.1, for any round $t$, Alice has $R_{1,t} + R_{2,t} = R_{3,t} + R_{4,t} = 0$, and Bob has $S_{1,t} + S_{2,t} = S_{3,t} + S_{4,t} = 0$.*

We now sketch the proof that consists of four steps. Informally, using lemmas 4.2 and 4.3, the first step says that the uninformative strategies $\text{opt}_3$ and $\text{opt}_4$ can not completely dominate other strategies. Specifically, both agents can not choose an option between $\text{opt}_3$ and $\text{opt}_4$ with a probability larger than $0.75$. With the first step, lemma 4.2 and lemma 4.3, we only need to focus on agents' reward of the truthtelling $\text{opt}_1$ shown in fig. 1. We partition the space into three types of events. *Good events* happen when $R_1, S_1$ are both very large or very small, *bad events* happen when one of $R_1, S_1$ is very large and one of them is very small, and *intermediate states* are the states between them. The second step removes the possibility that Alice and Bob continue using different reports $\text{opt}_1$ and $\text{opt}_2$. Therefore, we can always escape "bad events" and enter "intermediate states". The third step further shows that if the game is in an intermediate state, there exists a constant probability that the game will get into a good events that leads to truthful convergence in a constant number of rounds. Hence, after the game enters "good events", which leaves us the final step: showing their strategies converges to either both truth telling $\text{opt}_1$ or flipping $\text{opt}_2$ truthful convergence.

Now we discuss each steps in more details but defer all the proofs to appendix F.

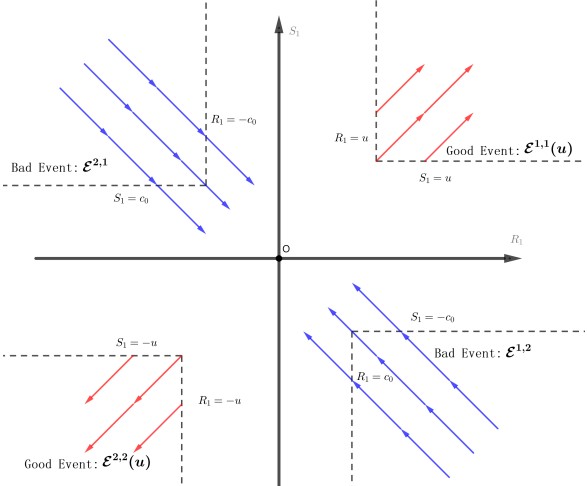

Figure 1: A schematic diagram of behaviors of $R_{1,t}$ and $S_{1,t}$.

**Step 1: Choosing $\text{opt}_1, \text{opt}_2$ with Nonzero Probability**

Combining Lemmas 4.2 and 4.3, we have the following Lemma 4.4, which completes step 1.

**Lemma 4.4.** *Given the game defined in theorem 4.1, for any round $t \geq 0$ and $i \in \{1, 2\}$, if $\sum_{j=1}^{t} r_{i,j} \geq 0$, the probability for Alice to choose $\text{opt}_i$ is larger than $\frac{1}{4}$; if $\sum_{j=1}^{t} s_{i,t} \geq 0$, the probability for Bob to choose $\text{opt}_i$ is larger than $\frac{1}{4}$.*

**Step 2: Escaping Bad Events**

Before introduce the formal statement of step 2, we define two "bad events". Given $c_0$ for all $t \geq 1$ $\mathcal{E}_t^{1,2} := \{R_{1,t} > c_0, \text{ and } S_{2,t} > c_0\}$ and $\mathcal{E}_t^{2,1} := \{R_{2,t} > c_0 \text{ and } S_{1,t} > c_0\}$. We will specify $c_0$ later. Intuitively, when $c_0$ is sufficiently large, $\mathcal{E}_t^{1,2}$ implies that Alice and Bob will choose $\text{opt}_1, \text{opt}_2$ respectively with a probability close to 1 in following rounds. For simplicity, we treat each event $\mathcal{E}$ as an indicator function, i.e., $\mathcal{E}$ happens if and only if $\mathcal{E} = 1$. In order to prove that these two bad events cannot go on forever, we want to show that when $\mathcal{E}_t^{1,2}$ happens, $R_{1,t} - R_{2,t}$ will tend to decrease at a rapid rate. Therefore, Alice will eventually deviate from $\text{opt}_1$ to choose $\text{opt}_2$ with a relatively high probability.

By assumption 3.5 and lemma 4.2, given $\delta > 0$ there exists a constant $c_1$ such that when $R_{1,t} > c_1$, Alice chooses $\text{opt}_1$ with probability larger than $1 - \delta$; when $S_{2,t} > c_1$, Bob chooses $\text{opt}_2$ with probability larger than $1 - \delta$. Let $\gamma_1 = P_{X,Y}(1,1) + P_{X,Y}(0,0) - P_{X,Y}(1,0) - P_{X,Y}(0,1)$ $\gamma_2 = (P_{X,Y}(1,1) - P_{X,Y}(0,0))^2 - (P_{X,Y}(1,0) - P_{X,Y}(0,1))^2$.

**Lemma 4.5.** *Given the game defined in theorem 4.1, there exists a $\delta > 0$ and corresponding $c_1$ such that for any round $t$, $\mathbb{E}[r_{1,t+1} - r_{2,t+1} | S_{1,t-1} > c_1 + 1] \geq \frac{\gamma_1 - \gamma_2}{2} > 0$.*

Given such $\delta$ and $c_1$ in lemma 4.5, we set $c_0 = c_1 + \lceil \frac{1000}{\gamma_1 - \gamma_2} \rceil + 1$. Because in each round $R_{1,t}, R_{2,t}$ and $S_{1,t}, S_{2,t}$ vary by at most 1, if $\mathcal{E}_t^{1,2}$ happens Alice chooses $\text{opt}_1$ and Bob chooses $\text{opt}_2$ with probability larger than $1 - \delta$ independently for the next $\lceil \frac{1000}{\gamma_1 - \gamma_2} \rceil + 1$ rounds. Similar argument holds for $\mathcal{E}_t^{2,1}$. We use the above observation to show lemma 4.6.

**Lemma 4.6.** *Given the game defined in theorem 4.1, for all $t$ and history $\mathcal{H}_t \in \mathcal{E}_t^{1,2}$, we have* $\mathbb{E}\left[ \sum_{j=1}^{\lceil \frac{1000}{\gamma_1 - \gamma_2} \rceil + 1} (r_{1,t+j} - r_{2,t+j}) \middle| \mathcal{H}_t \right] \leq -100.$

This lemma formalizes the blue arrows in Fig. 1. With this lemma, we get the main result of step two.

**Lemma 4.7.** *Given the game defined in theorem 4.1, $\Pr \left\{ \limsup_{t \to \infty} \overline{\mathcal{E}_t^{1,2} \vee \mathcal{E}_t^{2,1}} = 1 \right\} = 1$.*

If we treat $R_{1,t} - R_{2,t}$ as money of Alice, Lemma 4.7 is similar to the gambler's ruin problem [7]. More specifically, $R_{1,t} - R_{2,t}$ has a negative expected growth each $\lceil \frac{1000}{\gamma_1 - \gamma_2} \rceil + 1$ rounds by Lemma 4.6, so $R_{1,t} - R_{2,t}$ will always become small enough to escape $\mathcal{E}^{1,2}$.

**Step 3: From Intermediate States to Good Events**

We define a series of "good events" at first. For all $u \in \mathbb{N}^+$ and $t \geq 1$, $\mathcal{E}_t^{1,1}(u) := \{R_{1,t} \geq u, \text{ and } S_{1,t} \geq u\}$ and $\mathcal{E}_t^{2,2}(u) := \{R_{2,t} \geq u \text{ and } S_{2,t} \geq u\}$.

To the end, we want to show that $\vee_{t \in \mathbb{N}^+} \mathcal{E}_t^{1,1}(u)$ and $\vee_{t \in \mathbb{N}^+} \mathcal{E}_t^{2,2}(u)$ happen with probability 1 for any $u \in \mathbb{N}^+$. Formally, we claim Lemma 4.8.

**Lemma 4.8.** *Given the game defined in theorem 4.1, for all $u$ there exists $\lambda_u$ so that for any $T$ with history $\mathcal{H}_T \in \overline{\mathcal{E}_T^{1,2} \vee \mathcal{E}_T^{2,1}}$, we have* $\Pr \left\{ \left( \vee_{i=T}^{T+4(u+c_0)+100} \mathcal{E}_t^{1,1}(u) \right) \vee \left( \vee_{i=T}^{T+4(u+c_0)+100} \mathcal{E}_t^{2,2}(u) \right) = 1 \middle| \mathcal{H}_T \right\} \geq \lambda_u.$

This lemma generally says that when the agents are in an intermediate state that $\overline{\mathcal{E}_T^{1,2} \vee \mathcal{E}_T^{2,1}} = 1$, they have a constant probability $\lambda_u$ to get into a good event in the next $4(u + c_0) + 100$ rounds. In order to prove this lemma, we define a nice event $\mathcal{P}_T$ such that $\mathcal{P}_T$ happens with probability larger than $\lambda_u$ and $\mathcal{P}_T$ implies the good events happen in no more than $4(u + c_0) + 100$ rounds. Formally, event $\mathcal{P}_T$ is defined as the following: First, for $j = T, \ldots, T_1$ until some round $T_1 \geq T$ such that $S_{1,T_1} \geq 0$, $x_{j+1} = 1 - \hat{y}_j, y_{j+1} = 1 - \hat{x}_j$, Alice uses strategy $\text{opt}_1$, Bob uses strategy $\text{opt}_2$. Then, Alice and Bob uses strategy $\text{opt}_1$ for $4u + 50$ rounds and signals are generated as $x_{j+1} = y_{j+1} = -x_j$ for $j \geq T_1$. We can use Lemma 4.4 to prove that $\lambda_u = (\min_{i,j \in \{0,1\}} \{P_{X,Y}(i,j)\} \times 0.25^2)^{100+4(u+c_0)}$ is a feasible lower bound.

**Step 4: Good Events Lead to Truthful Convergence**

Similar to step 2, we have lemma 4.9 to show that $\mathcal{E}_t^{1,1}(u)$ can lead to increasing of $R_{1,t} - R_{2,t}$ at first. The idea is completely similar to Lemma 4.6 and it formalize the red arrows in Fig. 1.

**Lemma 4.9.** *Given the game defined in theorem 4.1, for all $u > 2c_0 + 1$ and $t$ if $\mathcal{H}_t \in \mathcal{E}_t^{1,1}\left(\lfloor \frac{u}{2} \rfloor\right)$, we have $\mathbb{E}\left[\sum_{j=1}^{\lceil \frac{1000}{\gamma_1 - \gamma_2} \rceil + 1} (r_{1,t+j} - r_{2,t+j}) \middle| \mathcal{H}_t\right] \geq 100$.*

Using Lemma 4.9, we are able to prove that for any $\varepsilon$, there exists a $u$ such that when $\mathcal{E}_t^{1,1}(u)$ or $\mathcal{E}_t^{2,2}(u)$ happens, Alice and Bob will tend to choose $(\text{opt}_1, \text{opt}_1)$ or $(\text{opt}_2, \text{opt}_2)$ with increasingly higher probability. Formally, we propose Lemma 4.10.

**Lemma 4.10.** *Given the game defined in theorem 4.1, for all $\epsilon > 0$ there exists $u \in \mathbb{N}^+$ such that given a history $\mathcal{H}_T \in \mathcal{E}_T^{1,1}(u) \vee \mathcal{E}_T^{2,2}(u)$, we have*
$$\Pr\left\{\forall i \in \mathbb{N}, \mathcal{E}_{T+\left(\lceil \frac{1000}{\gamma_1 - \gamma_2} \rceil + 1\right)i}^{1,1}\left(\lfloor \tfrac{u}{2} \rfloor + i\right) \vee \mathcal{E}_{T+\left(\lceil \frac{1000}{\gamma_1 - \gamma_2} \rceil + 1\right)i}^{2,2}\left(\lfloor \tfrac{u}{2} \rfloor + i\right) = 1 \middle| \mathcal{H}_T\right\} \geq 1 - \varepsilon.$$

We design a sub-martingale $\{D_i\}_{i \in \mathbb{N}}$ that is proportional to $R_{1, T+i\left(\lceil \frac{1000}{\gamma_1 - \gamma_2} \rceil + 1\right)} - R_{2, T+i\left(\lceil \frac{1000}{\gamma_1 - \gamma_2} \rceil + 1\right)}$, and use Azuma-Hoeffding inequality to prove lemma 4.10.

## 5 Simulations

We simulate the CA mechanism with various learning algorithms: the Hedge algorithms, follow the perturbed leader, follow the leader, and $\epsilon$-greedy, and repeat the process 400 times with 800 rounds on each algorithm each time. We define the *converge proportion* in round $t$ as the fraction of the simulations where both agents report truthfully $\text{opt}_1$ (or both use $\text{opt}_2$) in all the subsequent rounds.

In our simulations, we use the following private signal distribution that satisfies assumption 2.2: $P_{X,Y}(0,0) = P_{X,Y}(1,1) = 0.4, P_{X,Y}(1,0) = P_{X,Y}(0,1) = 0.2$. Moreover, Alice and Bob are using the same learning algorithms in our simulations that are listed below: First, Follow the Leader algorithm (FTL) chooses $\text{opt}_i$ with probability proportional to $\mathbb{I}[R_i = \max\{R_1, R_2, R_3, R_4\}]$. Follow the perturbed leader (FPL*, where * can be 1, 4 or 8) adds a uniform random noise between 0 and * and choose strategy $\text{opt}_i$ with probability $\Pr\left\{R_i + p_i = \max_{j \in [4]}\{R_j + p_j\} \middle| p_j \overset{\text{iid}}{\sim} \mathcal{U}[0, *]\right\}$. We consider FPL1, FPL4, and FPL8. Hedge algorithm 1 choose $i$ with probability proportional to $3^{R_i/2}$ that is an implementation by choosing $\epsilon = 0.5$ for the multiplicative weights algorithm of Arora et al. [2]. Hedge algorithm 2 chooses $i$ with probability proportional to $e^{R_i}$ that is an implementation by choosing $\beta = 1$ of exponentially weighted averaged forecaster introduced by Freund and Schapire [10]. Finally, $\epsilon$-greedy algorithm uses time varying $\epsilon = \frac{1}{(t+1)^2}$ at round $t$. Note that $\epsilon$-greedy is not in $\mathcal{A}$ but still achieves truthful convergence.

In Figure 2, all our algorithms converge to truth-telling. First, all reward-based online learning algorithms (the Hedge algorithms, follow the perturbed leader, and follow the leader) exhibit truthful convergence that aligns with our theoretical result, theorem 4.1. Moreover, although FTL is generally not no-regret, CA mechanism still works well with it. Additionally, we observe that when an algorithm explores less (e.g. FPL4 vs FPL8), it converges faster, but very little exploration does not further improve the convergence rate (e.g. FTL vs FPL2). Finally, we also find that the $\epsilon$-greedy with time decreasing $\epsilon$, which is not a reward-based online learning algorithm, also shows truthful convergence. This suggests that the CA mechanism may have truthful convergence beyond reward-based online learning algorithms.

## 6 Conclusions

In this paper, we study sequential peer prediction with learning agents and prove that the notion of no-regret alone is not sufficient for truthful convergence. We then define a family of reward-based learning algorithms and show that the CA mechanism is able to achieve truthful convergence when agents use algorithms in this family. Finally, we give a discussion on the converge rates of different learning agents based on simulations.

This is the first theoretical study on peer prediction with learning agents. There are many open problems and future directions to extend this work. We believe similar proof techniques can be used to extend our results to settings where agents' private signals are generated by a Markov chain with

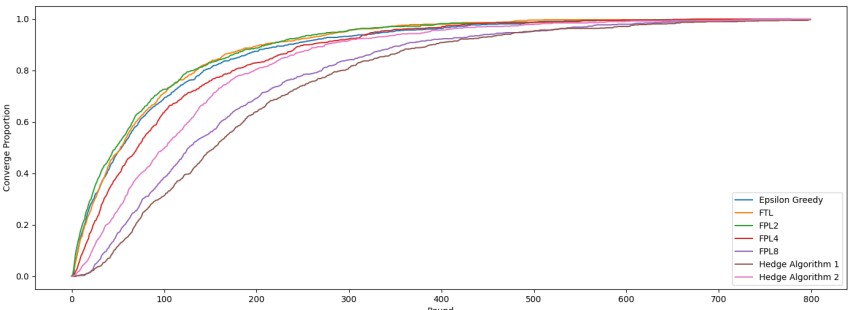

Figure 2: Convergence Rates of Learning Algorithms in CA.

some assumptions on the transition matrix. Moreover, this work is only restricted to binary signals and it is still an open problem whether there exists a mechanism for non-binary settings that can promise truthful convergence. For the learning agents, one could consider a more general family of learning algorithms such as when $f$ is time-varying.

## Acknowledgments and Disclosure of Funding

The authors would like to thank the anonymous reviewers for their valuable comments and constructive feedback. This work is partially supported by the National Science Foundation under Grant No. IIS 2007887 and by the National Science Foundation and Amazon under Grant No. FAI 2147187.

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
