# Appendix

## A  Basic Math

### A.1  Martingale and Concentration

In this section we will define martingales and some of its properties.

**Definition A.1** (Martingale). Let $\mathcal{F} = \{\mathcal{F}_t\}_{t \in \mathbb{N}}$ be a filtration, which is an increasing sequence of $\sigma$-field. A *martingale* with respect to $\mathcal{F}$ is a sequence $D_0, D_1, D_2, \cdots$ adapted to $\mathcal{F}$ ($D_t \in \mathcal{F}_t$ for all $t$) that satisfies for any time $t$,

$$\mathbb{E}\left[|D_t|\right] < \infty,$$
$$\mathbb{E}\left[D_{t+1}|\mathcal{F}_t\right] = D_t.$$

There are two extensions of a martingale that replace the equality of conditional probability by upper and lower bounds.

**Definition A.2** (Sub-martingale and super-martingale). Let $\mathcal{F} = \{\mathcal{F}_t\}_{t \in \mathbb{N}}$ be a filtration, which is an increasing sequence of $\sigma$-field. A *sub-martingale* with respect to $\mathcal{F}$ is a sequence $D_0, D_1, D_2, \cdots$ adapted to $\mathcal{F}$ that satisfies for any time $t$,

$$\mathbb{E}\left[|D_t|\right] < \infty,$$
$$\mathbb{E}\left[D_{t+1}|\mathcal{F}_t\right] \geq D_t.$$

A *super-martingale* with respect to $\mathcal{F}$ is a sequence $D_0, D_1, D_2, \cdots$ adapted to $\mathcal{F}$ that satisfies for any time $t$,

$$\mathbb{E}\left[|D_t|\right] < \infty,$$
$$\mathbb{E}\left[D_{t+1}|\mathcal{F}_t\right] \leq D_t.$$

For a sub-martingale (or super-martingale), we have the Azuma–Hoeffding inequality [1], which is a concentration result for the values of martingales.

**Theorem A.3** (Azuma–Hoeffding inequality). *Suppose $\{D_t\}_{t \in \mathbb{N}}$ is a martingale (or super-martingale) and $|D_t - D_{t-1}| \leq c_t$ almost surely. Then for any $T \in \mathbb{N}^+$ and $\epsilon \in \mathbb{R}^+$, we have*

$$\Pr\{D_t - D_0 \geq \epsilon\} \leq \exp\left(\frac{-\epsilon^2}{2\sum_{t=1}^{T} c_t^2}\right).$$

*And symmetrically, if $\{D_t\}_{t \in \mathbb{N}}$ is a sub-martingale, we have*

$$\Pr\{D_t - D_0 \leq -\epsilon\} \leq \exp\left(\frac{-\epsilon^2}{2\sum_{t=1}^{T} c_t^2}\right).$$

Now we define stopping time. This is intuitively a condition such that the "decision" whether to stop in the $t^{th}$ round should be based on information of the first $t$ rounds, instead of any future information.

**Definition A.4** (Stopping time). $\tau$ is called a *stopping time* for a filtration $\mathcal{F}$ if and only if $\{\tau = t\} \in \mathcal{F}_t, \forall t$.

### A.2  Limit Inferior and Limit Superior

Limit inferior and limit superior are defined on sequences, representing limit bounds of a sequence. To meet our needs, our definition focus on discrete metric.

**Definition A.5.** Let $\{\mathcal{E}_t\}_{t \in \mathbb{N}^+}$ be a sequence of events. Limit inferior and limit superior of this sequence are

$$\liminf_{t \to +\infty} \mathcal{E}_t = \vee_{t=1}^{+\infty} \left(\wedge_{i=t}^{+\infty} \mathcal{E}_i\right),$$

and

$$\limsup_{t \to +\infty} \mathcal{E}_t = \wedge_{t=1}^{+\infty} \left(\vee_{i=t}^{+\infty} \mathcal{E}_i\right).$$

### A.3 Borel-Cantelli Lemma

In this section, we formally introduce Borel-Cantelli lemma. Its proof can be found in [9].

**Theorem A.6** (Borel-Cantelli lemma). *Let $\mathcal{E}_1, \mathcal{E}_2, \cdots$ be a sequence of events in some probability space. Borel-Cantelli lemma states that if $\sum_{t=1}^{+\infty} \Pr\{\mathcal{E}_t\} < \infty$, then the probability that infinitely many of $\mathcal{E}_t$'s occur is $0$, or more strictly,*

$$\Pr\left\{\limsup_{t \to +\infty} \mathcal{E}_t\right\} = 0.$$

## B  Truthfulness of CA mechanism for Bayesian Agents

In this section, we assume that both agents are using consistent strategies as previous works assume [5, 25, 17, 23, 15, 28]. The formal definition of consistent strategy is given in definition B.1.

**Definition B.1** (Consistent strategy). In a repeated game, a strategy profile $\sigma$ is a *consistent strategy* if and only if for each agent $i$, she adopts $\sigma_i$ identically over each round of the game.

In a sequential peer prediction game, for agent $i$, we set the average payoffs of agent $i$ in $T$ rounds as her utility.

Then we introduce Bayesian Nash equilibrium. A Bayesian Nash equilibrium (BNE) is strategies of agents that have the maximal expected payoff for each player given their beliefs on environments and others' strategies [12].

**Definition B.2** (Bayesian Nash equilibiurm). A strategy profile $\sigma$ is a *Bayesian Nash equilibrium* if and only if for every agent $i$, the expected payoff of using $\sigma_i$ for agent $i$ is maximal keeping other agents' strategies unchanged. Moreover, A strategy profile $\sigma$ is a *strict Bayesian Nash equilibrium* if and only if for every agent $i$, the expected payoff of using $\sigma_i$ for agent $i$ is strictly larger than any other strategies keeping other agents' strategies unchanged.

Moreover, We say a peer prediction mechanism is *strongly truthful* if agents in truthtelling equilibrium get strictly higher payment than any other non-permutation equilibrium [25]. Here, a permutation equilibrium is the strategy profile that agents report a permutation of the signal. Formally, we have definition B.3.

**Definition B.3** (Strongly truthful). In a peer prediction game, if agents are using consistent strategies, a mechanism is *strongly truthful* if and only if truthtelling is a BNE and also guarantees larger agent welfare than any non-permutation equilibrium. Here, welfare is defined by each agent's expected payoff so that is to say, the expected payoff of each agent using truthtelling strategy profile is strictly higher than the expected payoff using non-permutation equilibrium.

Though Dasgupta and Ghosh [5], Shnayder et al. [25] have proved that CA mechanism is strongly truthful in non-sequential settings, our settings are slightly different from theirs and we rewrite the proof for binary sequential peer prediction settings. More specifically, our CA mechanism uses the last round agreement term instead of average agreement term in the payoffs and we are focusing on average payoffs instead of total payoffs. Before the complete proof, we have the following lemma.

**Lemma B.4.** *For binary sequential signal peer prediction games under assumption 2.1 and assumption 2.2, if both agents are using consistent strategies, CA mechanism renders truthtelling strategy profile a strict Bayesian Nash equilibrium.*

*Proof.* We know that Alice and Bob are Bayesian agents using consistent strategies. Then we can suppose when Alice gets signal 1, she reports 1 with probability $p_1$; when she gets signal 0, she reports 0 with probability $p_0$. Similarly, Bob is using a fixed strategy that when he gets signal 1, he reports 1 with probability $q_1$; when he gets signal 0, he reports 0 with probability $q_0$. Therefore, we can compute the first term in payoff of Alice and Bob (see eq. (1)), $\mathbb{E}[\mathbb{I}[\hat{x}_t = \hat{y}_t]]$, by

$$\mathbb{E}[\mathbb{I}[\hat{x}_t = \hat{y}_t]] = P_{X,Y}(1,1)(p_1 q_1 + (1-p_1)(1-q_1)) + P_{X,Y}(0,0)(p_0 q_0 + (1-p_0)(1-q_0))$$
$$+ P_{X,Y}(1,0)(p_1(1-q_0) + (1-p_1)q_0) + P_{X,Y}(0,1)(q_1(1-p_0) + (1-q_1)p_0).$$

Also, we can compute the second term

$$\mathbb{E}[\mathbb{I}[\hat{x}_t = \hat{y}_{t-1}]] = (P_{X,Y}(1,1) + P_{X,Y}(1,0))(P_{X,Y}(0,1) + P_{X,Y}(1,1))(p_1 q_1 + (1-p_1)(1-q_1))$$
$$+ (P_{X,Y}(0,0) + P_{X,Y}(0,1))(P_{X,Y}(1,0) + P_{X,Y}(0,0))(p_0 q_0 + (1-p_0)(1-q_0))$$
$$+ (P_{X,Y}(1,1) + P_{X,Y}(1,0))(P_{X,Y}(0,0) + P_{X,Y}(1,0))(p_1(1-q_0) + (1-p_1)q_0)$$
$$+ (P_{X,Y}(0,1) + P_{X,Y}(0,0))(P_{X,Y}(0,1) + P_{X,Y}(1,1))((1-p_0)q_1 + p_0(1-q_1)).$$

Similarly, $\mathbb{E}[\mathbb{I}[\hat{y}_t = \hat{x}_{t-1}]]$ also equals to this expression.

To prove BNE, we only need to prove that when $q_0 = q_1 = 1$, we have

$$\underset{(p_0, p_1)}{\arg\max} \left\{ \mathbb{E}[\mathbb{I}[\hat{x}_t = \hat{y}_t]] - \mathbb{E}[\mathbb{I}[\hat{x}_t = \hat{y}_{t-1}]] \right\} = (1,1). \tag{2}$$

The other side when $p_0 = p_1 = 1$ is symmetrical.

Actually, we have

$$\frac{\partial(\mathbb{E}[\mathbb{I}[\hat{x}_t = \hat{y}_t] | q_0 = q_1 = 1] - \mathbb{E}[\mathbb{I}[\hat{x}_t = \hat{y}_{t-1}] | q_0 = q_1 = 1])}{\partial p_0}$$
$$= P_{X,Y}(0,0) - P_{X,Y}(0,1)$$
$$+ (P_{X,Y}(0,0) + P_{X,Y}(0,1))(P_{X,Y}(0,1) + P_{X,Y}(1,1) - P_{X,Y}(0,0) - P_{X,Y}(1,0))$$
$$= 2P_{X,Y}(0,0) - 2(P_{X,Y}(0,0) + P_{X,Y}(0,1))(P_{X,Y}(0,0) + P_{X,Y}(1,0))$$
$$= 2(P_{X,Y}(0,0)P_{X,Y}(1,1) - P_{X,Y}(0,1)P_{X,Y}(1,0))$$
$$> 0,$$

and

$$\frac{\partial(\mathbb{E}[\mathbb{I}[\hat{x}_t = \hat{y}_t] | q_0 = q_1 = 1] - \mathbb{E}[\mathbb{I}[\hat{x}_t = \hat{y}_{t-1}] | q_0 = q_1 = 1])}{\partial p_1}$$
$$= P_{X,Y}(1,1) - P_{X,Y}(1,0)$$
$$+ (P_{X,Y}(1,1) + P_{X,Y}(1,0))(P_{X,Y}(0,0) + P_{X,Y}(1,0) - P_{X,Y}(1,1) - P_{X,Y}(0,1))$$
$$= 2P_{X,Y}(1,1) - 2(P_{X,Y}(1,1) + P_{X,Y}(1,0))(P_{X,Y}(1,1) + P_{X,Y}(0,1))$$
$$= 2(P_{X,Y}(1,1)P_{X,Y}(0,0) - P_{X,Y}(0,1)P_{X,Y}(1,0))$$
$$> 0,$$

which indicate eq. (2). $\qquad \square$

Furthermore, we can find all the Nash equilibria for Bayesian agents under our CA mechanism in sequential peer prediction setting. We have theorem B.5.

**Theorem B.5.** *For binary sequential signal peer prediction games under assumption 2.1 and assumption 2.2, if CA mechanism is used and agents adopt consistent strategies, there are three types of Nash equilibria for agents, which are*

1. *truth-telling $opt_1$,*

2. *flip the signal $opt_2$,*

3. *report regardless of private signals (uninformative reports).*

*Proof.* Using the same notations in the proof of lemma B.4, we deduce that

$$\frac{\partial(\mathbb{E}[\mathbb{I}[\hat{x}_t = \hat{y}_t]] - \mathbb{E}[\mathbb{I}[\hat{x}_t = \hat{y}_{t-1}]])}{\partial p_0}$$
$$= (P_{X,Y}(1,1)P_{X,Y}(0,0) - P_{X,Y}(1,0)P_{X,Y}(0,1))(2q_0 + 2q_1)$$
$$- 2(P_{X,Y}(1,1)P_{X,Y}(0,0) - P_{X,Y}(1,0)P_{X,Y}(0,1)),$$

and similarly,

$$\frac{\partial(\mathbb{E}[\mathbb{I}[\hat{x}_t = \hat{y}_t]] - \mathbb{E}[\mathbb{I}[\hat{x}_t = \hat{y}_{t-1}]])}{\partial p_1}$$
$$= (P_{X,Y}(1,1)P_{X,Y}(0,0) - P_{X,Y}(1,0)P_{X,Y}(0,1))(2q_0 + 2q_1)$$
$$- (P_{X,Y}(1,1)P_{X,Y}(0,0) - P_{X,Y}(1,0)P_{X,Y}(0,1)).$$

Therefore, we know that when $q_0 + q_1 > 1$, the best response of Alice is $p_0 = p_1 = 1$; when $q_0 + q_1 < 1$, the best response of Alice is $p_0 = p_1 = 0$. Symmetrically, we can deduce that when $p_0 + p_1 > 1$, the best response of Alice is $q_0 = q_1 = 1$; when $p_0 + p_1 < 1$, the best response of Alice is $q_0 = q_1 = 0$.

Let $p_0^*, p_1^*, q_0^*, q_1^*$ be a Nash equilibrium for this peer prediction game under CA mechanism. Then given $p_0^*, p_1^*, q_0^*, q_1^*$ should be one of the best responses of Bob; given $q_0^*, q_1^*, p_0^*, p_1^*$ should be one of the best responses of Alice.

If $p_0^* + p_1^* > 1$, then the only best response of Bob is $q_0 = q_1 = 1$, so $q_0^* = q_1^* = 1$. When $q_0^* = q_1^* = 1$, the only best response of Alice is $p_0 = p_1 = 1$, so we can deduce that $p_0^* = p_1^* = q_0^* = q_1^* = 1$ is the unique Nash equilibrium for this case.

If $p_0^* + p_1^* < 1$, then the only best response of Bob is $q_0 = q_1 = 0$, so $q_0^* = q_1^* = 0$. When $q_0^* = q_1^* = 0$, the only best response of Alice is $p_0 = p_1 = 0$, so we can deduce that $p_0^* = p_1^* = q_0^* = q_1^* = 0$ is the unique Nash equilibrium for this case.

Symmetrically, when $q_0^* + q_1^* \neq 1$, there are still only these two Nash equilibria. Moreover, it is easy to verify that $p_0^*, p_1^*, q_0^*, q_1^*$ that satisfies $p_0^* + p_1^* = q_0^* + q_1^* = 1$ is Nash equilibrium. Therefore, all the uninformative reports are Nash equilibria. These are exactly what we want to prove. $\square$

From the proof of lemma B.4, we can observe that when both agents use $\mathrm{opt}_1$, their expected payoffs are both

$$
\begin{aligned}
&P_{X,Y}(1,1) + P_{X,Y}(0,0) - (P_{X,Y}(1,1) + P_{X,Y}(1,0))(P_{X,Y}(0,1) + P_{X,Y}(1,1)) \\
&- (P_{X,Y}(0,1) + P_{X,Y}(0,0))(P_{X,Y}(1,0) + P_{X,Y}(0,0)) \\
&= 2P_{X,Y}(1,1)P_{X,Y}(0,0) - 2P_{X,Y}(1,0)P_{X,Y}(0,1) \\
&> 0.
\end{aligned}
$$

Also, we can observe that when both agents use uninformative reports, their expected payoffs are both $0$. According to theorem B.5, we know that the only non-permutation equilibrium is uninformative reports. Therefore, we deduce that CA mechanism is strongly truthful.

**Theorem B.6.** *For binary sequential signal peer prediction games under assumption 2.1 and assumption 2.2, if agents adopt consistent strategies, CA mechanism is strongly truthful.*

## C  Proofs and Details of Section 3.1

### C.1  Impossibility of Truthful Convergence for No Regret Agents

*Proof of theorem 3.1.* First, for any sequential information elicitation mechanism because Alice and Bob can use arbitrary no regret algorithm, there exist no-regret for Alice and Bob. If the truth-telling has regret on $\mathcal{M}$, the statement trivially holds. Otherwise, suppose truth-telling is no regret on $\mathcal{M}$. We have for all $T$ the expectations $\mu_T^X = \max_i \sum_{t \leq T} \mathbb{E} r_{i,t} - \sum_{t \leq T} \mathbb{E} r_t$ and $\mu_T^Y = \max_i \sum_{t \leq T} \mathbb{E} s_{i,t} - \sum_{t \leq T} \mathbb{E} s_t$ satisfy

$$
\mu_T^X \text{ and } \mu_T^Y = o(T). \tag{3}
$$

We will use probabilistic method to show the existence of a deterministic and no regret sequence of strategies $(\mathrm{opt}_t^X, \mathrm{opt}_t^Y)_{t=1,\dots}$ that consists of reporting $1$ and $0$ regardless of private signal $\mathrm{opt}_3, \mathrm{opt}_4$. As a result, when Alice and Bob use the sequence, the algorithm is no regret but $\mathcal{M}$ does not achieve truthful convergence on such algorithm. To find such sequence, it is sufficient for us to find a deterministic sequence $(\hat{x}_t^*, \hat{y}_t^*)_{t \geq 1}$ so that for all $T$

$$
\max_i \sum r_{i,t}^* - \sum r_t^* \text{ and } \max_i \sum s_{i,t}^* - \sum s_t^* = o(T), \tag{4}
$$

because we can define an online learning algorithms so that Alice play $\mathrm{opt}_t^X = \mathrm{opt}_{2+\hat{x}_t^*}$ and Bob play $\mathrm{opt}_t^Y = \mathrm{opt}_{2+\hat{y}_t^*}$ for all $t$. Additionally, if we can find $(\hat{x}_t^*, \hat{y}_t^*)_{1 \leq t}$ and $T_0$ that for all $T \geq T_0$,

$$
|\max_i \sum r_{i,t} - \sum r_t - \mu_T^X| \text{ and } |\max_i \sum s_{i,t} - \sum s_t - \mu_T^Y| \leq T^{2/3}, \tag{5}
$$

by eq. (3) $(\hat{x}_t^*, \hat{y}_t^*)_{1 \leq t}$ satisfies eq. (4).

Because their signal mutually independent across different rounds and each round's signals can only affect at most $2k - 1$ rounds of payoff, each pair of signals only changes $\max_i \sum_{t \leq T} r_{i,t} - \sum_{t \leq T} r_t$ by $2(2k - 1) \leq 4k$. Therefore, by Chernoff bound using method of bounded difference, we have for all $T$ and $\epsilon > 0$

$$\Pr\left[\left|\max_i \sum_{t \leq T} r_{i,t} - \sum_{t \leq T} r_t - \mu_T^X\right| \leq \epsilon T\right] \leq 2\exp\left(-\frac{2\epsilon^2}{16k^2}T\right)$$

$$\Pr\left[\left|\max_i \sum_{t \leq T} s_{i,t} - \sum_{t \leq T} s_t - \mu_T^Y\right| \leq \epsilon T\right] \leq 2\exp\left(-\frac{2\epsilon^2}{16k^2}T\right)$$

Thus, we can take $\epsilon = T^{-1/3}$ and $T_0$ large enough and prove the random payoffs of truth-telling satisfy eq. (5) for all $T \geq T_0$ with high probability by union bound. Therefore, by probabilistic method there exists a (determistic) sequence $(\hat{\mathbf{x}}_t^*, \hat{\mathbf{y}}_t^*)_{1 \leq t}$ so that eq. (5) holds for all $T \geq T_0$. $\qquad\square$

### C.2 Truthful Convergence Implies No regret

*Proof of theorem 3.2.* Given reports $\hat{\mathbf{x}}$ and signals $\mathbf{x}$, let $\hat{\mathbf{x}}_{t-k:t} = (\hat{x}_{t-k+1}, \ldots, \hat{x}_t)$ be a slice of reports $\hat{\mathbf{x}}$, and $\text{opt}_i(\mathbf{x}_{t-k:t}) = (\text{opt}_i(x_{t-k+1}), \ldots, \text{opt}_i(x_t))$ be a slice reports under consistent strategy $\text{opt}_i$. Given $T$, we define four functions on the signals

$$F_T^i(\mathbf{x}, \mathbf{y}) := \sum_{t=0}^{T} \bar{M}^X(\text{opt}_i(\mathbf{x}_{t-k:t}), \mathbf{y}_{t-k:t}) - \bar{M}^X(\mathbf{x}_{t-k:t}, \mathbf{y}_{t-k:t})$$

for $i = 1, 2, 3, 4$. We want to show the value of $F_T^i$ is small. Specifically, we bound the probability of the following good event

$$\mathcal{G} := \{\max_i F_T^i(\mathbf{x}, \mathbf{y}) \leq T^{2/3}\}.$$

First because $\bar{M}$ is strongly truthful, $\mathbb{E}_{\sigma^X}[\bar{M}^X(\hat{\mathbf{x}}, \mathbf{y})] \leq \mathbb{E}[\bar{M}^X(\mathbf{x}, \mathbf{y})]$ and $\mathbb{E}_{\sigma^Y}[\bar{M}^Y(\mathbf{x}, \hat{\mathbf{y}})] \leq \mathbb{E}[\bar{M}^Y(\mathbf{x}, \mathbf{y})]$, the expectation is non-positive

$$\mathbb{E}[F_T^i(\mathbf{x}, \mathbf{y})] = \sum_{t=0}^{T} \mathbb{E}_{\sigma^X = \text{opt}_i}\left[\bar{M}^X(\hat{\mathbf{x}}_{t-k:t}, \mathbf{y}_{t-k:t}) - \bar{M}^X(\mathbf{x}_{t-k:t}, \mathbf{y}_{t-k:t})\right] \leq 0$$

for all $i$. Second, because each round's signals are mutually independent and can only affect at most $2k - 1$ round of payoff, by Chernoff bound on $F_T^i$, we have for all $i$, $\Pr\left[F_T^i(\mathbf{x}, \mathbf{y}) \geq T^{2/3}\right] \leq \Pr\left[F_T^i(\mathbf{x}, \mathbf{y}) \geq T^{2/3} + \mathbb{E}F_T^i\right] \leq \exp\left(-\frac{2}{16k^2}T^{1/3}\right)$. By union bound, we have

$$\Pr[\mathcal{G}] \geq 1 - 4\exp\left(-\frac{2}{16k^2}T^{1/3}\right). \tag{6}$$

On the other hand, the truthful convergence consists of two disjoint events: both converging to truth telling $\text{opt}_1$, and both converging to the flipping strategy $\text{opt}_2$. By symmetric suppose the first event happens with a nonzero probability

$$\mathcal{E} := \{\lim_{t \to +\infty} \text{opt}_t^X = \lim_{t \to +\infty} \text{opt}_t^Y = \text{opt}_1\}.$$

Then there exists a random round $t^*$ so that $\text{opt}_t^X = \text{opt}_t^Y = \text{opt}_1$ for all $t \geq t^*$ given $\mathcal{E}$. To bound the expected regret conditional on $\mathcal{E}$, we consider two cases: If the converge time $t^*$ is greater than $T^{2/3}$, we use the truthful convergence to show the probability is small. Otherwise if the converge time $t^*$ is smaller than $T^{2/3}$, we can ignore the first term.

Formally, Alice's expected regret is

$$\mathbb{E}\left[Reg^X(T) \mid \mathcal{E}\right] = \mathbb{E}\left[\mathbf{1}[t^* > T^{2/3}]Reg^X(T) \mid \mathcal{E}\right] + \mathbb{E}\left[\mathbf{1}[t^* \leq T^{2/3}]Reg^X(T) \mid \mathcal{E}\right].$$

For the first term, $\mathbb{E}\left[\mathbf{1}[t^* > T^{2/3}]Reg^X(T) \mid \mathcal{E}\right] \leq T\mathbb{E}\left[\mathbf{1}[t^* > T^{2/3}] \mid \mathcal{E}\right] = T\Pr[t^* > T^{2/3} \mid \mathcal{E}]$. Because $\Pr[t^* > T^{2/3}] = o(1)$ as $T$ increases due to truthful convergence, and $\mathcal{E}$ happens with nonzero probability, we have

$$\mathbb{E}\left[\mathbf{1}[t^* > T^{2/3}]Reg^X(T) \mid \mathcal{E}\right] = o(T). \tag{7}$$

On the other hand, when $t^* \leq T^{2/3}$ happens,

$$Reg^X(T) = \max_i \sum_{t \leq T} r_{i,t} - \sum_{t \leq T} r_t$$

$$\leq (t^* + k) + \max_i \sum_{t=t^*+k}^{T} r_{i,t} - r_t \qquad (r_t \text{ and } r_{i,t} \text{ are in } [0,1])$$

$$= (t^* + k) + \max_i \sum_{t=t^*+k}^{T} \bar{M}^X(\text{opt}_i(\mathbf{x}_{t-k:t}), \mathbf{y}_{t-k:t}) - \bar{M}^X(\mathbf{x}_{t-k:t}, \mathbf{y}_{t-k:t}) \qquad (\mathcal{E} \text{ happens})$$

$$\leq 2(t^* + k) + \max_i F_T^i(\mathbf{x}, \mathbf{y}) \qquad (\text{adding addition } t^* + k \text{ terms})$$

$$\leq 2(T^{2/3} + k) + \max_i F_T^i(\mathbf{x}, \mathbf{y}) \qquad (t^* \leq T^{2/3})$$

Therefore,

$$\mathbb{E}\left[\mathbf{1}[t^* \leq T^{2/3}]Reg^X(T) \mid \mathcal{E}\right] \leq 2(T^{2/3} + k) + \mathbb{E}\left[\max_i F_T^i(\mathbf{x}, \mathbf{y}) \mid \mathcal{E}\right]. \qquad (8)$$

To bound the second term, we partition the expectation by whether $\mathcal{G}$ happens or not

$$\mathbb{E}\left[\max_i F_T^i(\mathbf{x}, \mathbf{y}) \mid \mathcal{E}\right]$$

$$= \mathbb{E}\left[\max_i F_T^i(\mathbf{x}, \mathbf{y}) \mid \mathcal{E}, \mathcal{G}\right] \Pr[\mathcal{G} \mid \mathcal{E}] + \mathbb{E}\left[\max_i F_T^i(\mathbf{x}, \mathbf{y}) \mid \mathcal{E}, \neg\mathcal{G}\right] \Pr[\neg\mathcal{G} \mid \mathcal{E}]$$

$$\leq T^{2/3} \Pr[\mathcal{G} \mid \mathcal{E}] + T \Pr[\neg\mathcal{G} \mid \mathcal{E}] \qquad (\text{definition of } \mathcal{G})$$

$$\leq T^{2/3} + T \frac{\Pr[\neg\mathcal{G}]}{\Pr[\mathcal{E}]} = O(T^{2/3}) \qquad (\text{by eq. (6)})$$

Therefore, with eqs. (7) and (8) we show $\mathbb{E}\left[Reg^X(T) \mid \mathcal{E}\right] = o(T) + 2(T^{2/3} + k) + O(T^{2/3}) = o(T)$ that completes the proof. □

## D   Justifications of Reward-Based Online Learning Algorithm Family $\mathcal{A}$

In this section, we have two subsections to give justifications for learning algorithm family $\mathcal{A}$. In the first part, we introduce two common used learning algorithms and show that they are both reward-based online learning algorithms. The second part gives justifications for assumption 3.5.

### D.1   Learning Algorithms in $\mathcal{A}$

In this section, we do not focus on peer prediction problems but consider learning algorithms used on general online decision problems.

#### D.1.1   Follow the Perturbed Leader

FPL algorithm is designed by [13]. In their work, they have proved that FPL algorithm achieves no best-in-hindsight regret in full-information online decision problems. For simplicity, we consider FPL using on online decision problem with four options $\text{opt}_i, i \in [4]$ in total. The algorithm let the agent choose an arbitrary option among $\arg\max_{\text{opt}_i \in \{\text{opt}_1, \text{opt}_2, \text{opt}_3, \text{opt}_4\}} (R_{i,t-1} + p_{i,t})$ in the $t^{th}$ round of the game. Here, $p_{i,t}$'s are i.i.d. sampled from a particular noise distribution $\mathcal{N}$. Because of variety of the noise distribution, FPL algorithm actually contains a large family of learning algorithms, i.e., Hannan's algorithm [11] and Follow the Leader algorithm (FTL).

In formal, we have the following theorem.

**Theorem D.1.** *Algorithm 1 is a reward-based online learning algorithm included in $\mathcal{A}$.*

*Proof.* To prove that FPL algorithm is in our algorithm family $\mathcal{A}$, we need to design a function $f^{\text{FPL}}$ such that

$$f_i^{\text{FPL}}(R_1, R_2, R_3, R_4) = \Pr\left\{ R_i + p_i = \max_{j \in [4]} \{R_j + p_j\} \,\middle|\, p_j \overset{\text{iid}}{\sim} \mathcal{N}, j \in [4] \right\}$$

---

**ALGORITHM 1:** FPL algorithm.

---
**Input:** Noise distribution $\mathcal{N}$.

1  **for** $t = 1, 2, \cdots$ **do**
2       For each option $\text{opt}_i, i \in [4]$, sample $p_{i,t}$ independently from noise distribution $\mathcal{N}$.
3       Arbitrarily choose an option among $\arg\max_{\text{opt}_i \in \{\text{opt}_1, \text{opt}_2, \text{opt}_3, \text{opt}_4\}} (R_{i,t-1} + p_{i,t})$ in the $t^{th}$ round.
4  **end**

---

for $i \in [4]$. Using this function based on cumulative payoffs, $A_{f^{\text{FPL}}}$ is equivalent to algorithm 1. In detail, the probability of choosing $\text{opt}_i$ in the $t^{th}$ round in algorithm 1 is exactly

$$\Pr\left\{ R_{i,t-1} + p_{i,t} = \max_{j \in [4]}\{R_{j,t-1} + p_{j,t}\} \,\middle|\, p_{j,t} \overset{\text{iid}}{\sim} \mathcal{N}, j \in [4] \right\}.$$

Now we only need to verify that $f^{\text{FPL}}$ satisfies assumption 3.3, assumption 3.5 and assumption 3.4.

Assumption 3.3 holds because the expression of probability

$$\Pr\left\{ R_i + p_i = \max_{j \in [4]}\{R_j + p_j\} \,\middle|\, p_j \overset{\text{iid}}{\sim} \mathcal{N}, j \in [4] \right\}$$

is symmetrical with respect to $i$.

We know that for $\forall \epsilon > 0$, we can find a large enough positive constant $n$ such that $\Pr\{p - p' \geq n | p, p' \overset{\text{iid}}{\sim} \mathcal{N}\} < \epsilon$. Therefore, when $R_1 > \max\{R_2, R_3, R_4\} + n$, we can deduce that

$$
\begin{aligned}
&f_1^{\text{FPL}}(R_1, R_2, R_3, R_4) \\
&= \Pr\left\{ R_1 + p_1 = \max_{i \in [4]}\{R_i + p_i\} \,\middle|\, p_i \overset{\text{iid}}{\sim} \mathcal{N}, i \in [4] \right\} \\
&\geq 1 - \sum_{i=2,3,4} \Pr\left\{ R_1 + p_1 < R_i + p_i \,\middle|\, p_i \overset{\text{iid}}{\sim} \mathcal{N}, i \in [4] \right\} \\
&\geq 1 - \sum_{i=2,3,4} \Pr\left\{ p_1 + n < p_i \,\middle|\, p_i \overset{\text{iid}}{\sim} \mathcal{N}, i \in [4] \right\} \\
&\geq 1 - 3\epsilon.
\end{aligned}
$$

Therefore, Assumption 3.5 holds. About Assumption 3.4, it is obvious by the definition of $f^{\text{FPL}}$ because each of $p_i$'s are sampled from an identical noise distribution. $\qquad\square$

### D.1.2  Multiplicative Weight Algorithm

Multiplicative weights algorithm (or hedge algorithm) is first introduced in [10], which is called *exponentially weighted averaged forecaster* by them. It is also a no-regret algorithm in full-information online decision problem. It can be written as Algorithm 2 for an online decision problem with four options $\text{opt}_i, i \in [4]$.

---

**ALGORITHM 2:** Multiplicative Weights algorithm in [10].

---
**Input:** A positive constant $\beta$.

1  Initialize $w_{i,1} = 1$ for $i \in [4]$.
2  **for** $t = 1, 2, \cdots$ **do**
3       Choose option $\text{opt}_i$ with probability $q_{i,t} = \frac{w_{i,t}}{\sum_{j \in [4]} w_{j,t}}$ in the $t^{th}$ round.
4       Update $w_{i,t+1} = w_{i,t} \exp(\beta r_{i,t})$.
5  **end**

---

In formal, we have the following theorem.

**Theorem D.2.** *Algorithm 2 is a reward-based online learning algorithm included in $\mathcal{A}$.*

*Proof.* To prove that algorithm 2 is in our algorithm family $\mathcal{A}$, we can use a function $f^{\text{hedge 2}}$ such that

$$f_i^{\text{hedge 2}}(R_1, R_2, R_3, R_4) = \frac{e^{\beta R_i}}{e^{\beta R_1} + e^{\beta R_2} + e^{\beta R_3} + e^{\beta R_4}},$$

for $i \in [4]$. Using this function based on cumulative payoffs, $A_{f^{\text{hedge 2}}}$ is equivalent to algorithm 2. In detail, the probability of choosing $\text{opt}_i$ in the $t^{th}$ round in algorithm 2 is exactly

$$\frac{w_i}{w_1 + w_2 + w_3 + w_4} = \frac{\prod_{j=1}^{t-1} \exp(\beta r_{i,t})}{\sum_{k \in [4]} \prod_{j=1}^{t-1} \exp(\beta r_{k,t})}$$

$$= \frac{e^{\beta R_{i,t-1}}}{e^{\beta R_{1,t-1}} + e^{\beta R_{2,t-1}} + e^{\beta R_{3,t-1}} + e^{\beta R_{4,t-1}}}.$$

Now we only need to verify that $f^{\text{hedge 2}}$ satisfies assumption 3.3, assumption 3.5 and assumption 3.4.

Assumption 3.3 holds because $\frac{e^{\beta R_i}}{e^{\beta R_1} + e^{\beta R_2} + e^{\beta R_3} + e^{\beta R_4}}$ is symmetrical with respect to $R_1, R_2, R_3, R_4$. Assumption 3.4 holds because $\beta > 0$ and exponential function is monotonic. When $R_1 - \max\{R_2, R_3, R_4\} > n > 0$, we have

$$f_1^{\text{hedge 2}}(R_1, R_2, R_3, R_4) = \frac{e^{\beta R_1}}{e^{\beta R_1} + e^{\beta R_2} + e^{\beta R_3} + e^{\beta R_4}}$$

$$= \frac{1}{1 + e^{\beta(R_2 - R_1)} + e^{\beta(R_3 - R_1)} + e^{\beta(R_4 - R_1)}}$$

$$> \frac{1}{1 + 3e^{-\beta n}}.$$

Therefore, we deduce that

$$\lim_{R_1 - \max\{R_2, R_3, R_4\} \to +\infty} f_1(R_1, R_2, R_3, R_4) \geq \lim_{R_1 - \max\{R_2, R_3, R_4\} \to +\infty} \frac{1}{1 + 3e^{-\beta n}} = 1.$$

Therefore, assumption 3.5 holds. $\square$

## D.2 Justifications for Assumption 3.5

In this section, we prove that assumption 3.5 is a necessary condition for a reward-based online learning algorithm to be no-regret.

**Theorem D.3.** *For a reward-based function $f : \mathbb{R}^4 \to \triangle^3$, if*

$$\lim_{R_1 - \max\{R_2, R_3, R_4\}} f_1(R_1, R_2, R_3, R_4) = 1 - c < 1,$$

*mechanism $A_f$ cannot be a no-regret algorithm for general online decision problem.*

*Proof.* We design an online decision problem such that $r_{1,t} = 2, r_{2,t} = r_{3,t} = r_{4,t} = 1$ for $t \in \mathbb{N}^+$. According to the description of $f$, there exists a $n$ such that when $R_1 - \max\{R_2, R_3, R_4\} \geq n$, $f_1(R_1, R_2, R_3, R_4) < 1 - \frac{c}{2}$. Notice that when $t > n + 1$, we have $R_{1,t-1} = 2t > t + n = \max\{R_{2,t-1}, R_{3,t-1}, R_{4,t-1}\} + n$, therefore, when $t > n + 1$, the agent chooses $\text{opt}_1$ with probability at most $1 - \frac{c}{2}$ in the $t^{th}$ round.

Therefore, we can deduce that when $T > n + 2$, $\mathbb{E}[Reg(T)] \geq \frac{c}{2}(T - n - 1)$, which is a linear function of $T$. Therefore, $A_f$ is not no-regret. $\square$

# E   More Algorithms in $\mathcal{A}$ Applying on CA Mechanism Binary Sequential Peer Prediction

In appendix D.1, we have introduced two widely used algorithms that are reward-based online learning algorithms. In this section, we show that there are even more existing learning algorithms contained by $\mathcal{A}$ when the game is exactly binary sequential peer prediction using CA mechanism.

## E.1 Replicator Dynamics

Replicator dynamics track a set of agents in a repeating game and each agent chooses a pure strategy with a probability proportional to expected payoffs deviating to higher-payoff options. We use a similar implementation as [26] in a general discrete form here to show that replicator dynamics are also in $\mathcal{A}$ for the binary signal peer prediction problem. To be more specific, during the repeating game, the agent maintains four probabilities $q_{i,t}, i \in [4]$ and choose $\text{opt}_i$ with probability $q_{i,t}$, and then update them in the end of the $t^{th}$ round. The updating rule is set as below:

$$q_{i,t+1} = \frac{h(r_{i,t})q_{i,t}}{\sum_{j \in [4]} h(r_{j,t})q_{j,t}}, i \in [4], \tag{9}$$

where $h : \mathbb{R} \to \mathbb{R}^+$ is a monotonic function. Due to the variety of $h$, our discretized replicator dynamics contain extensive learning algorithms. Common discretization of replicator dynamics set $h$ as exponential function or linear function, but we consider general $h$ functions here.

---

**ALGORITHM 3:** Replicator dynamics.

---

**Input:** Monotonic function $h : \mathbb{R} \to \mathbb{R}^+$.

1 **for** $t = 1, 2, \cdots$ **do**

2      Choose option $\text{opt}_i$ with probability $q_{i,t}$ in the $t^{th}$ round.

3      Set $q_{i,t+1} = \frac{h(r_{i,t})q_{i,t}}{\sum_{j \in [4]} h(r_{j,t})q_{j,t}}, i \in [4]$.

4 **end**

---

We have the following theorem.

**Theorem E.1.** *When applying on binary sequential peer prediction using CA mechanism, algorithm 3 is included in $\mathcal{A}$.*

*Proof.* According to eq. (9), we know that

$$\frac{q_{1,t+1}}{q_{2,t+1}} = \prod_{i=1}^{t} \frac{h(r_{1,i})}{h(r_{2,i})}$$

$$= \left(\frac{h(1)}{h(-1)}\right)^{\sum_{i \in [t]} \mathbb{I}[r_{1,i}=1 \wedge r_{2,i}=-1]} \times \left(\frac{h(-1)}{h(1)}\right)^{\sum_{i \in [t]} \mathbb{I}[r_{1,i}=-1 \wedge r_{2,i}=1]} \tag{10}$$

$$= \left(\frac{h(1)}{h(-1)}\right)^{\frac{R_{1,t}-R_{2,t}}{2}}. \tag{11}$$

Here, eq. (10) and eq. (11) are because there are only three realizations of $(r_{1,i}, r_{2,i})$, which are $(1, -1), (0, 0)$ and $(-1, 1)$ by the definition of CA mechanism and binary sequential peer prediction. Similarly for $\text{opt}_3, \text{opt}_4$, we can deduce that $\frac{q_{3,t+1}}{q_{4,t+1}} = \left(\frac{h(1)}{h(-1)}\right)^{\frac{R_{3,t}-R_{4,t}}{2}}$. Moreover, we have

$$\frac{q_{1,t+1}q_{2,t+1}}{q_{3,t+1}q_{4,t+1}} = \frac{h(r_{1,t})h(r_{2,t})}{h(r_{3,t})h(r_{4,t})} \frac{q_{1,t}q_{2,t}}{q_{3,t}q_{4,t}}$$

$$= \frac{q_{1,t}q_{2,t}}{q_{3,t}q_{4,t}} = \frac{q_{1,t-1}q_{2,t-1}}{q_{3,t-1}q_{4,t-1}} = \cdots = 1.$$

This is because $(r_{1,i}, r_{2,i}, r_{3,i}, r_{4,i})$ has only five possible realizations, which are $(0, 0, 0, 0)$, $(1, -1, 1, -1), (1, -1, -1, 1), (-1, 1, 1, -1)$ and $(-1, 1, -1, 1)$ according to the definition of CA mechanism and binary sequential peer prediction for any $i \in \mathbb{N}^+$.

According to lemma 4.3, we know that $R_{1,t} + R_{2,t} = R_{3,t} + R_{4,t}$. Therefore, we can deduce that for replicator dynamics applying on binary sequential peer prediction using CA mechanism, we have

$$q_{1,t+1} = \frac{\left(\frac{h(1)}{h(-1)}\right)^{\frac{R_{1,t}}{2}}}{\left(\frac{h(1)}{h(-1)}\right)^{\frac{R_{1,t}}{2}} + \left(\frac{h(1)}{h(-1)}\right)^{\frac{R_{2,t}}{2}} + \left(\frac{h(1)}{h(-1)}\right)^{\frac{R_{3,t}}{2}} + \left(\frac{h(1)}{h(-1)}\right)^{\frac{R_{4,t}}{2}}},$$

$$q_{2,t+1} = \frac{\left(\frac{h(1)}{h(-1)}\right)^{\frac{R_{2,t}}{2}}}{\left(\frac{h(1)}{h(-1)}\right)^{\frac{R_{1,t}}{2}} + \left(\frac{h(1)}{h(-1)}\right)^{\frac{R_{2,t}}{2}} + \left(\frac{h(1)}{h(-1)}\right)^{\frac{R_{3,t}}{2}} + \left(\frac{h(1)}{h(-1)}\right)^{\frac{R_{4,t}}{2}}},$$

$$q_{3,t+1} = \frac{\left(\frac{h(1)}{h(-1)}\right)^{\frac{R_{3,t}}{2}}}{\left(\frac{h(1)}{h(-1)}\right)^{\frac{R_{1,t}}{2}} + \left(\frac{h(1)}{h(-1)}\right)^{\frac{R_{2,t}}{2}} + \left(\frac{h(1)}{h(-1)}\right)^{\frac{R_{3,t}}{2}} + \left(\frac{h(1)}{h(-1)}\right)^{\frac{R_{4,t}}{2}}},$$

$$q_{4,t+1} = \frac{\left(\frac{h(1)}{h(-1)}\right)^{\frac{R_{4,t}}{2}}}{\left(\frac{h(1)}{h(-1)}\right)^{\frac{R_{1,t}}{2}} + \left(\frac{h(1)}{h(-1)}\right)^{\frac{R_{2,t}}{2}} + \left(\frac{h(1)}{h(-1)}\right)^{\frac{R_{3,t}}{2}} + \left(\frac{h(1)}{h(-1)}\right)^{\frac{R_{4,t}}{2}}}.$$

Hence, now replicator dynamics behave the same as mechanism $A_{f^{\text{replicator}}}$ such that

$$f_1^{\text{replicator}}(R_1, R_2, R_3, R_4) = \frac{\left(\frac{h(1)}{h(-1)}\right)^{\frac{R_1}{2}}}{\left(\frac{h(1)}{h(-1)}\right)^{\frac{R_1}{2}} + \left(\frac{h(1)}{h(-1)}\right)^{\frac{R_2}{2}} + \left(\frac{h(1)}{h(-1)}\right)^{\frac{R_3}{2}} + \left(\frac{h(1)}{h(-1)}\right)^{\frac{R_4}{2}}},$$

$$f_2^{\text{replicator}}(R_1, R_2, R_3, R_4) = \frac{\left(\frac{h(1)}{h(-1)}\right)^{\frac{R_2}{2}}}{\left(\frac{h(1)}{h(-1)}\right)^{\frac{R_1}{2}} + \left(\frac{h(1)}{h(-1)}\right)^{\frac{R_2}{2}} + \left(\frac{h(1)}{h(-1)}\right)^{\frac{R_3}{2}} + \left(\frac{h(1)}{h(-1)}\right)^{\frac{R_4}{2}}},$$

$$f_3^{\text{replicator}}(R_1, R_2, R_3, R_4) = \frac{\left(\frac{h(1)}{h(-1)}\right)^{\frac{R_3}{2}}}{\left(\frac{h(1)}{h(-1)}\right)^{\frac{R_1}{2}} + \left(\frac{h(1)}{h(-1)}\right)^{\frac{R_2}{2}} + \left(\frac{h(1)}{h(-1)}\right)^{\frac{R_3}{2}} + \left(\frac{h(1)}{h(-1)}\right)^{\frac{R_4}{2}}},$$

$$f_4^{\text{replicator}}(R_1, R_2, R_3, R_4) = \frac{\left(\frac{h(1)}{h(-1)}\right)^{\frac{R_4}{2}}}{\left(\frac{h(1)}{h(-1)}\right)^{\frac{R_1}{2}} + \left(\frac{h(1)}{h(-1)}\right)^{\frac{R_2}{2}} + \left(\frac{h(1)}{h(-1)}\right)^{\frac{R_3}{2}} + \left(\frac{h(1)}{h(-1)}\right)^{\frac{R_4}{2}}}.$$

It is easy to verify that $f$ satisfies assumption 3.3, assumption 3.5 and assumption 3.4.

In detail, assumption 3.3 holds because $f^{\text{replicator}}$ is symmetrical obviously with respect to $R_1, R_2, R_3, R_4$. Moreover, we know that

$$f_1^{\text{replicator}}(R_1, R_2, R_3, R_4) \geq \frac{\left(\frac{h(1)}{h(-1)}\right)^{\frac{R_1}{2}}}{\left(\frac{h(1)}{h(-1)}\right)^{\frac{R_1}{2}} + 3\left(\frac{h(1)}{h(-1)}\right)^{\frac{\max\{R_2,R_3,R_4\}}{2}}}$$

$$= \frac{1}{1 + \left(\frac{h(1)}{h(-1)}\right)^{\frac{\max\{R_2,R_3,R_4\}-R_1}{2}}}.$$

Therefore, when $R_1 - \max\{R_2, R_3, R_4\} \to +\infty$, we have $\frac{1}{1+\left(\frac{h(1)}{h(-1)}\right)^{\frac{\max\{R_2,R_3,R_4\}-R_1}{2}}} \to 1$. Thus

$f_1^{\text{replicator dynamics}}(R_1, R_2, R_3, R_4) \to 1$. Hence, we have proved that assumption 3.5 holds. Finally, assumption 3.4 also holds obviously according to the definition of $f^{\text{replicator}}$ and monotonicity of exponential function. $\qquad\square$

## E.2 An Alternating Version of Multiplicative Weights Algorithm

In appendix D.1, we have already introduced an exponential updating function form of multiplicative weights algorithm [10]. There is another version multiplicative weights algorithm introduced in the survey [2], which can be written as algorithm 4 for our particular binary sequential peer prediction problem using CA mechanism.

---

**ALGORITHM 4:** Multiplicative Weights algorithm in [2].

**Input:** A positive constant $\beta$.

1   Initialize $w_{i,1} = 1$ for $i \in [4]$.

2   **for** $t = 1, 2, \cdots$ **do**

3      Choose option $\text{opt}_i$ with probability $q_{i,t} = \frac{w_{i,t}}{\sum_{j \in [4]} w_{j,t}}$ in the $t^{th}$ round.

4      Update $w_{i,t+1} = \begin{cases} (1+\beta)w_{i,t} & r_{i,t} = 1 \\ w_{i,t} & r_{i,t} = 0 \\ (1-\beta)w_{i,t} & r_{i,t} = -1 \end{cases}$ .

5   **end**

---

We only need to set $h : \mathbb{R} \to \mathbb{R}^+$ to satisfy that $h(1) = 1 + \beta$, $h(0) = 1$ and $h(-1) = 1 - \beta$. Then algorithm 3 behaves completely the same as algorithm 4. Therefore, according to theorem E.1, we have theorem E.2 for this alternating version of multiplicative weights algorithm.

**Theorem E.2.** *When applying on binary sequential peer prediction using CA mechanism, algorithm 4 is included in $\mathcal{A}$.*

# F   Proofs and Details of Section 4

## F.1   Proofs of Properties of CA Mechanism in Binary Signal Peer Prediction Games

### F.1.1   Proof of Lemma 4.2

*Proof.* By the definition of CA mechanism, we know that $r_{3,j} = \mathbb{I}[1 = \hat{y}_j] - \mathbb{I}[1 = \hat{y}_{j-1}]$ for $j \in \mathbb{N}^+$. Therefore, we can deduce that

$$
\begin{aligned}
R_{3,t} &= \sum_{j=1}^{t} r_{3,j} \\
&= \sum_{j=1}^{t} \left( \mathbb{I}[1 = \hat{y}_j] - \mathbb{I}[1 = \hat{y}_{j-1}] \right) \\
&= \sum_{j=1}^{t} \mathbb{I}[1 = \hat{y}_j] - \sum_{j=0}^{t-1} \mathbb{I}[1 = \hat{y}_j] \\
&= \mathbb{I}[1 = \hat{y}_t] - \mathbb{I}[1 = \hat{y}_0] \in [-1, 1].
\end{aligned}
$$

For strategy $\sum_{j=1}^{t} p_{4,j}, \sum_{j=1}^{t} q_{3,j}, \sum_{j=1}^{t} q_{4,j}$, the deductions are all similar.      $\square$

### F.1.2 Proof of Lemma 4.3

*Proof.* By the definition of CA mechanism, we know that $r_{1,j} = \mathbb{I}[x_j = \hat{y}_j] - \mathbb{I}[x_j = \hat{y}_{j-1}]$ and $r_{2,j} = \mathbb{I}[1 - x_j = \hat{y}_j] - \mathbb{I}[1 - x_j = \hat{y}_{j-1}]$. Therefore, we can deduce that

$$
\begin{aligned}
R_{1,t} + R_{2,t} &= \sum_{j=1}^{t} (r_{1,j} + r_{2,j}) \\
&= \sum_{j=1}^{t} (\mathbb{I}[x_j = \hat{y}_j] - \mathbb{I}[x_j = \hat{y}_{j-1}]) + \sum_{j=1}^{t} (\mathbb{I}[1 - x_j = \hat{y}_j] - \mathbb{I}[1 - x_j = \hat{y}_{j-1}]) \\
&= \sum_{j=1}^{t} (\mathbb{I}[x_j = \hat{y}_j] + \mathbb{I}[1 - x_j = \hat{y}_j]) - \sum_{j=1}^{t} (\mathbb{I}[x_j = \hat{y}_{j-1}] + \mathbb{I}[1 - x_j = \hat{y}_{j-1}]) \\
&= t - t = 0.
\end{aligned}
$$

Similar deductions can be made for $R_{3,t} + R_{4,t}$, $S_{1,t} + S_{2,t}$ and $S_{3,t} + S_{4,t}$, which completes the proof. $\qquad\square$

### F.2 Proofs of Truthful Convergence for CA mechanism on Reward-based Algorithms

### F.2.1 Proof of Lemma 4.4

*Proof.* Because Alice and Bob are symmetric, we only consider Alice. By lemma 4.3, without loss of generality, we suppose that $\sum_{j=1}^{t} r_{1,j} \geq 0$. Then $R_{1,t} \geq 1$ and $R_{2,t} \leq 0$ or $R_{1,t} = R_{2,t} = 0$ because each $r_{1,j}$ equals to $-1, 0$ or $1$, which is an integer. If $R_{1,t} \geq 1$, then we can deduce that $R_{1,t} \geq \max_{i=2,3,4} R_{i,t}$. Therefore, $f_1(R_{1,t}, R_{2,t}, R_{3,t}, R_{4,t}) \geq \max_{i=2,3,4} f_i(R_{t,1}, R_{2,t}, R_{3,t}, R_{4,t})$ according to assumption 3.4. We know that $\sum_{i\in[4]} f_i(R_{1,t}, R_{2,t}, R_{3,t}, R_{4,t}) = 1$, so $f_1(R_{1,t}, R_{2,t}, R_{3,t}, R_{4,t}) \geq \frac{1}{4}$.

If $R_{1,t} = R_{2,t} = R_{3,t} = R_{4,t} = 0$, we can also deduce that $R_{1,t} \geq \max_{i=2,3,4} R_{i,t}$. Therefore, $f_1(R_{1,t}, R_{2,t}, R_{3,t}, R_{4,t}) \geq \max_{i=2,3,4} f_i(R_{t,1}, R_{2,t}, R_{3,t}, R_{4,t})$ according to assumption 3.4. We know that $\sum_{i\in[4]} f_i(R_{1,t}, R_{2,t}, R_{3,t}, R_{4,t}) = 1$, so $f_1(R_{1,t}, R_{2,t}, R_{3,t}, R_{4,t}) \geq \frac{1}{4}$.

Finally, we prove that $\{R_{1,t} = R_{2,t} = 0, R_{3,t} = 1, R_{4,t} = -1\}$ and $\{R_{1,t} = R_{2,t} = 0, R_{3,t} = -1, R_{4,t} = 1\}$ never occur in our game. According to the definition of CA mechanism, we know that $(r_{1,t}, r_{2,t}, r_{3,t}, r_{4,t})$ has only five possibilities, which are $(0,0,0,0)$, $(1,-1,1,-1)$, $(1,-1,-1,1)$, $(-1,1,1,-1)$ and $(-1,1,-1,1)$. Therefore, we can deduce that $r_{1,t} - r_{2,t} + r_{3,t} - r_{4,t}$ can be divided by 4. Hence, $R_{1,t} - R_{2,t} + R_{3,t} - R_{4,t} = \sum_{i=1}^{t} (r_{1,t} - r_{2,t} + r_{3,t} - r_{4,t})$ is also divided by 4. This indicates that both $\{\sum_{j=1}^{t} r_{1,j} = \sum_{j=1}^{t} r_{2,j} = 0, \sum_{j=1}^{t} r_{3,j} = 1, \sum_{j=1}^{t} r_{4,j} = -1\}$ and $\{\sum_{j=1}^{t} r_{1,j} = \sum_{j=1}^{t} r_{2,j} = 0, \sum_{j=1}^{t} r_{3,j} = -1, \sum_{j=1}^{t} r_{4,j} = 1\}$ cannot happen.

To sum up, the probability that Alice chooses $\mathrm{opt}_1$ is larger than $\frac{1}{4}$, which is what we want. $\qquad\square$

### F.2.2 Proof of Lemma 4.5

*Proof.* It is easy to verify that $\gamma_1 > 0$. Then under the situation that $S_{1,t-1} > c_1 + 1$, for the next round, the expectation of $r_{1,t+1} - r_{2,t+1}$ can be bounded as

$$
\begin{aligned}
\mathbb{E}[r_{1,t+1} - r_{2,t+1}] &= \mathbb{E}\left[ (\mathbb{I}[x_{t+1} = \hat{y}_{t+1}] - \mathbb{I}[x_{t+1} = \hat{y}_t]) - (\mathbb{I}[1 - x_{t+1} = \hat{y}_{t+1}] - \mathbb{I}[1 - x_{t+1} = \hat{y}_t]) \right] \\
&= \mathbb{E}\left[ \mathbb{I}[x_{t+1} = \hat{y}_{t+1}] - \mathbb{I}[1 - x_{t+1} = \hat{y}_{t+1}] \right] - \mathbb{E}\left[ \mathbb{I}[x_{t+1} = \hat{y}_t] - \mathbb{I}[1 - x_{t+1} = \hat{y}_t] \right] \\
&\geq \Pr\{\mathrm{opt}_{t+1}^Y = \mathrm{opt}_1\}\gamma_1 - (1 - \Pr\{\mathrm{opt}_{t+1}^Y = \mathrm{opt}_1\}) \\
&\quad - \Pr\{\mathrm{opt}_t^Y = \mathrm{opt}_1\}\gamma_2 + (1 - \Pr\{\mathrm{opt}_t^Y = \mathrm{opt}_1\}) \\
&> (1 - \delta)\gamma_1 - \delta - \max\{\gamma_2, (1 - \delta)\gamma_2 - \delta\}.
\end{aligned}
$$

We know that

$$\gamma_1 - \gamma_2 = (P_{X,Y}(1,1) + P_{X,Y}(0,0) - P_{X,Y}(1,0) - P_{X,Y}(0,1) - |P_{X,Y}(1,0) - P_{X,Y}(0,1)|)$$
$$+ (|P_{X,Y}(0,1) - P_{X,Y}(1,0)| - (P_{X,Y}(0,1) - P_{X,Y}(1,0))^2) + (P_{X,Y}(1,1) - P_{X,Y}(0,0))^2$$
$$> |P_{X,Y}(0,1) - P_{X,Y}(1,0)| - (P_{X,Y}(0,1) - P_{X,Y}(1,0))^2$$
$$> 0.$$

Therefore, we can always find $\delta$ such that

$$\mathbb{E}\left[r_{1,t+1} - r_{2,t+1}|S_{1,t-1} > c_1 + 1\right] \geq (1-\delta)\gamma_1 - \delta - \max\{\gamma_2, (1-\delta)\gamma_2 - \delta\} \geq \frac{\gamma_1 - \gamma_2}{2} > 0,$$

which is what we want. $\qquad\square$

### F.2.3   Proof of Lemma 4.6

*Proof.* Noticing that when $\mathcal{E}_t^{1,2}$ happens, Bob will choose $\text{opt}_2$ with probability larger than $1 - \delta$ in the next $\lceil \frac{1000}{\gamma_1 - \gamma_2} \rceil + 1$ rounds. Therefore, we can deduce that

$$\mathbb{E}\left[\sum_{j=1}^{\lceil \frac{1000}{\gamma_1-\gamma_2}\rceil+1} (r_{1,t+j} - r_{2,t+j}) \middle| \mathcal{H}_t \text{ such that } \mathcal{E}_t^{1,2} = 1\right]$$

$$= \sum_{j=1}^{\lceil \frac{1000}{\gamma_1-\gamma_2}\rceil+1} \left(\mathbb{E}\left[\mathbb{I}[x_{t+j} = \hat{y}_{t+j}] - \mathbb{I}[1 - x_{t+j} = \hat{y}_{t+j}] \middle| \mathcal{H}_t \text{ such that } \mathcal{E}_t^{1,2} = 1\right]\right.$$

$$\left. - \mathbb{E}\left[\mathbb{I}[x_{t+j} = \hat{y}_{t+j-1}] - \mathbb{I}[1 - x_{t+j} = \hat{y}_{t+j-1}] \middle| \mathcal{H}_t \text{ such that } \mathcal{E}_t^{1,2} = 1\right]\right)$$

$$\leq 2 + \sum_{j=2}^{\lceil \frac{1000}{\gamma_1-\gamma_2}\rceil+1} (\delta - (1-\delta)\gamma_1 + \max\{\gamma_2, (1-\delta)\gamma_2 - \delta\}) \qquad (12)$$

$$\leq 2 - \sum_{j=2}^{\lceil \frac{1000}{\gamma_1-\gamma_2}\rceil+1} \frac{\gamma_1 - \gamma_2}{2} \qquad (13)$$

$$< -100.$$

Here, eq. (12) is because when $j \geq 2$, we have

$$\mathbb{E}\left[\mathbb{I}[x_{t+j} = \hat{y}_{t+j}] - \mathbb{I}[1 - x_{t+j} = \hat{y}_{t+j}] \middle| \mathcal{H}_t \text{ such that } \mathcal{E}_t^{1,2} = 1\right]$$

$$- \mathbb{E}\left[\mathbb{I}[x_{t+j} = \hat{y}_{t+j-1}] - \mathbb{I}[1 - x_{t+j} = \hat{y}_{t+j-1}] \middle| \mathcal{H}_t \text{ such that } \mathcal{E}_t^{1,2} = 1\right]$$

$$\leq (1 - \Pr\{\text{opt}_{t+j}^Y = \text{opt}_1\}) - \Pr\{\text{opt}_{t+j}^Y = \text{opt}_1\}\gamma_1$$

$$- (1 - \Pr\{\text{opt}_{t+j-1}^Y = \text{opt}_1\}) + \Pr\{\text{opt}_{t+j-1}^Y = \text{opt}_1\}\gamma_2$$

$$< \delta - (1-\delta)\gamma_1 + \max\{\gamma_2, (1-\delta)\gamma_2 - \delta\}.$$

Moreover, eq. (13) holds according to definition of $\delta$ in lemma 4.5. $\qquad\square$

### F.2.4   Proof of Lemma 4.7

*Proof.* First we show the complement of event $\limsup_{t\to\infty} \overline{\mathcal{E}_t^{1,2} \vee \mathcal{E}_t^{2,1}}$ is

$$\liminf_{t\to\infty} \mathcal{E}_t^{1,2} \vee \liminf_{t\to\infty} \mathcal{E}_t^{2,1}. \qquad (14)$$

According to the definition of $\limsup$, we can write $\limsup_{t\to\infty} \overline{\mathcal{E}_t^{1,2} \vee \mathcal{E}_t^{2,1}} = 1$ as $\wedge_{t=1}^{\infty}\left(\vee_{i=t}^{\infty}\overline{\mathcal{E}_i^{1,2} \vee \mathcal{E}_i^{2,1}}\right) = 1$. This is equivalent to $\vee_{t=1}^{\infty}\left(\wedge_{i=t}^{\infty}\left(\mathcal{E}_i^{1,2} \vee \mathcal{E}_i^{2,1}\right)\right) = 0$ by De Morgan's law, which can be written as $\liminf_{t\to\infty}\left(\mathcal{E}_t^{1,2} \vee \mathcal{E}_t^{2,1}\right) = 0$.

More concretely, according to the definition of $\liminf$, $\limsup_{t\to\infty} \overline{\mathcal{E}_t^{1,2} \vee \mathcal{E}_t^{2,1}} = 1$ is

$$\left\{ \exists T \in \mathbb{N}^+, \forall t \geq T, \mathcal{E}_t^{1,2} \vee \mathcal{E}_t^{2,1} = 1 \right\}.$$

Therefore, we only need to prove that $\Pr\left\{ \exists T \in \mathbb{N}^+, \forall t \geq T, \mathcal{E}_t^{1,2} \vee \mathcal{E}_t^{2,1} = 1 \right\} = 0$ that is the complement of eq. (14). We denote all the game history before the $t^{th}$ round as $\mathcal{H}_t$. Then we only need to prove that for any $\mathcal{H}_T$, the conditional probability $\Pr\left\{ \mathcal{E}_t^{1,2} \vee \mathcal{E}_t^{2,1} = 1, \forall t \geq T \Big| \mathcal{H}_T \right\} = 0$. This is because

$$\Pr\left\{ \exists T \in \mathbb{N}^+, \forall t \geq T, \mathcal{E}_t^{1,2} \vee \mathcal{E}_t^{2,1} = 1 \right\} \leq \sum_{T=1}^{\infty} \sum_{\mathcal{H}_T} \Pr\{\mathcal{H}_t\} \Pr\left\{ \forall t \geq T, \mathcal{E}_t^{1,2} \vee \mathcal{E}_t^{2,1} = 1 \Big| \mathcal{H}_T \right\},$$

where the number of summed terms is countable and we know that the sum of countable infinite zeros is still zero.

Moreover, if $\mathcal{E}_t^{1,2} = 1$, $\mathcal{E}_{t+1}^{2,1} \neq 1$ according to the definition, we know that $\left\{ \mathcal{E}_t^{1,2} = 1, \mathcal{E}_{t+1}^{2,1} = 1 \right\}$ and $\left\{ \mathcal{E}_t^{2,1} = 1, \mathcal{E}_{t+1}^{1,2} = 1 \right\}$ always equals to zero. Therefore, $\left\{ \forall t \geq T, \mathcal{E}_t^{1,2} \vee \mathcal{E}_t^{2,1} = 1 \Big| \mathcal{H}_T \right\}$ is equivalent to

$$\left\{ \forall t \geq T, \mathcal{E}_t^{1,2} \vee \mathcal{E}_t^{2,1} = 1 \Big| \mathcal{H}_T \right\} \wedge \left( \wedge_{t=1}^{\infty} \overline{\left\{ \mathcal{E}_t^{1,2} = 1, \mathcal{E}_{t+1}^{2,1} = 1 \right\}} \right) \wedge \left( \wedge_{t=1}^{\infty} \overline{\left\{ \mathcal{E}_t^{2,1} = 1, \mathcal{E}_{t+1}^{1,2} = 1 \right\}} \right). \quad (15)$$

Moreover, suppose the event expressed as eq. (15) happens, we can deduce that for $t \geq T$, if $\mathcal{E}_t^{1,2} = 1$, $\mathcal{E}_{t+1}^{2,1} = 0$ because $\left\{ \mathcal{E}_t^{1,2} = 1, \mathcal{E}_{t+1}^{2,1} = 1 \right\} = 0$, so $\mathcal{E}_{t+1}^{1,2}$ still happens; if $\mathcal{E}_t^{2,1} = 1$, $\mathcal{E}_{t+1}^{1,2} = 0$ because $\left\{ \mathcal{E}_t^{2,1} = 1, \mathcal{E}_{t+1}^{1,2} = 1 \right\} = 0$, so $\mathcal{E}_{t+1}^{2,1}$ still happens. Thus, event in eq. (15) leads to $\left\{ \forall t \geq T, \mathcal{E}_t^{1,2} = 1 \Big| \mathcal{H}_T \right\} \vee \left\{ \forall t \geq T, \mathcal{E}_t^{2,1} = 1 \Big| \mathcal{H}_T \right\}$. Conversely, $\left\{ \forall t \geq T, \mathcal{E}_t^{1,2} = 1 \Big| \mathcal{H}_T \right\} \vee \left\{ \forall t \geq T, \mathcal{E}_t^{2,1} = 1 \Big| \mathcal{H}_T \right\}$ is obviously included in the event expressed as eq. (15). Hence, event $\left\{ \forall t \geq T, \mathcal{E}_t^{1,2} \vee \mathcal{E}_t^{2,1} = 1 \Big| \mathcal{H}_T \right\}$ is equivalent to

$$\left\{ \forall t \geq T, \mathcal{E}_t^{1,2} = 1 \Big| \mathcal{H}_T \right\} \vee \left\{ \forall t \geq T, \mathcal{E}_t^{2,1} = 1 | \mathcal{H}_T \right\}.$$

Therefore, without loss of generality, we only need to prove that $\Pr\left\{ \forall t \geq T, \mathcal{E}_t^{1,2} = 1 \Big| \mathcal{H}_T \right\} = 0$ for any game history $\mathcal{H}_T$. We show this by using Borel-Cantelli lemma (theorem A.6) and Azuma-Hoeffding inequality (theorem A.3) to find a sub-sequence of tasks $\left\{ t_i := T + i \left( \lceil \frac{1000}{\gamma_1 - \gamma_2} \rceil \right) : i \geq 1 \right\}$ for some $T$ so that $\mathcal{E}_{t_i}^{1,2}$ only happens finitely often. For simplicity, we denote $\lceil \frac{1000}{\gamma_1 - \gamma_2} \rceil$ by $\zeta$.

Let $\tau$ be a stopping time such that $\tau = \min_{t > T}\{R_{1,t} \leq c_0 \text{ or } S_{2,t} \leq c_0\}$ and $i_\tau = \lceil \frac{\tau - T}{\zeta + 1} \rceil$.

We design a series of new variables $\{D_i\}_{i=0,1,2,\ldots}$, where

$$D_i = \left( R_{1,T+i(\zeta+1)} - R_{2,T+i(\zeta+1)} + 100i \right) \mathbb{I}\left[ T + (i-1)(\zeta+1) < \tau \right]$$
$$+ \left( R_{1,T+i_\tau(\zeta+1)} - R_{2,T+i_\tau(\zeta+1)} + 100i_\tau \right) \mathbb{I}\left[ T + (i-1)(\zeta+1) \geq \tau \right].$$

Now we show that $\{D_i\}_{i=0,1,2,\ldots}$ is a super-martingale with bounded difference, and we will use Azuma-Hoeffding inequality (theorem A.3) to show the value of $D_i$ (and thus $R_{1,t}$) cannot be too big. Therefore, $\mathcal{E}_t^{1,2}$ can only happen finitely many times by Borel-Cantelli lemma (theorem A.6).

Actually, if $\mathcal{H}_{T+i(\zeta+1)}$ satisfies that $\mathbb{I}[T + i(\zeta+1) < \tau] = 1$, we have $\mathcal{E}_{T+i(\zeta+1)}^{1,2} = 1$. According to lemma 4.6, we have

$$\mathbb{E}\left[ D_{i+1} \Big| \mathcal{H}_{T+i(\zeta+1)} \right]$$
$$= \mathbb{E}\left[ \sum_{j=1}^{\zeta+1} (r_{1,T+i(\zeta+1)+j} - r_{2,T+i(\zeta+1)+j}) \Bigg| \mathcal{H}_{T+i(\zeta+1)} \right] + D_i + 100$$
$$\leq -100 + D_i + 100 = D_i.$$

On the other hand, if $\mathcal{H}_{T+i(\zeta+1)}$ satisfies that $\mathbb{I}[T + i\,(\zeta + 1) < \tau] = 0$, we know that $D_{i+1} = D_i$.

Moreover, when $i \leq i_\tau$, we know that $|D_i - D_{i-1}|$ is bounded by $102 + 2\zeta$ for any $i \in \mathbb{N}^+$ because $|R_{1,t+j} - R_{1,t}|$ is bounded by $j$ by definition for any $t, j \in \mathbb{N}$. When $i > i_\tau$, we know that $D_i = D_{i-1}$.

It is worthy to notice that $\mathcal{E}^{1,2}_{T+i(\zeta+1)} = 1$ implies $R_{1,T+i(\zeta+1)} > c_0$. Therefore, if $\mathcal{E}^{1,2}_{T+j} = 1$ for all $j > T$, $\tau$ does not exist, so we have $D_i > 2c_0 + 100i$. However, according to Azuma–Hoeffding inequality (theorem A.3), for $i > \frac{R_{1,T} - R_{2,T} - 2c_0}{100}$, we have

$$\Pr\{D_i > 2c_0 + 100i\} = \Pr\{D_i - D_0 > 2c_0 + 100i - (R_{1,T} - R_{2,T})\}$$
$$\leq \exp\left(\frac{-(2c_0 + 100i - (R_{1,T} - R_{2,T}))^2}{2i(102 + 2\zeta)^2}\right).$$

This upper bound of $\Pr\{D_i > 2c_0 + 100i\}$ decays exponentially in $i$. By the Borel-Cantelli lemma (theorem A.6), the event $\{R_{1,T+i(\zeta+1)} > c_0\}$ will occur only finitely often almost surely. This is contradictory to $\mathcal{E}^{1,2}_{T+j} = 1$ for all $j > T$, so $\Pr\{\mathcal{E}^{1,2}_t = 1, \forall t \geq T|\mathcal{H}_T\} = 0$. $\qquad\square$

### F.2.5 Proof of Lemma 4.8

*Proof.* By symmetry, we only consider the case that $\mathcal{H}_T$ with $R_{1,T} + S_{1,T} \geq 0$ and $R_{1,T} \geq S_{1,T}$. If $R_{1,T} \leq c_0$, then we have $S_{1,T} \geq -R_{1,T} \geq -c_0$; if $R_{1,T} > c_0$, because $\mathcal{E}^{1,2}_T \vee \mathcal{E}^{2,1}_T = 1$, $S_{2,T} \leq c_0$, so $S_{1,T} = -S_{2,T} \geq -c_0$. Therefore, we have $R_{1,T} \geq 0$ together with $S_{1,T} \geq -c_0$.

We now propose a process $\mathcal{P}_T$ with less than $T + 4(u + c_0) + 100$ of rounds, and we will prove that it happens with probability no less than a constant $\lambda_u$ given any $\mathcal{H}_T$ as we suppose. The process $\mathcal{P}_T$ is defined as

1. If $S_{1,T} \geq 0$, skip this phase. Otherwise, $x_{j+1} = 1 - \hat{y}_j, y_{j+1} = 1 - \hat{x}_j$ from $j = T$, Alice uses strategy opt$_1$, Bob uses strategy opt$_2$ until some round $T_1$ such that $S_{1,T_1} \geq 0$.

2. Alice and Bob uses strategy opt$_1$ for $4u + 50$ rounds and signals are generated as $x_{j+1} = y_{j+1} = -x_j$ for $j \geq T_1$.

Firstly, we prove that $\mathcal{P}_T$ will stop in $T + 4(u + c_0) + 100$ rounds and in the round $T_2$ exactly after $\mathcal{P}_T$, the game will enter good events such that $\mathcal{E}^{1,1}(u) \vee \mathcal{E}^{2,2}(u) = 1$. This part can be proved simply according to the definition of $\mathcal{P}_T$.

During this process, we claim that $R_{1,i}$ and $S_{1,i}$ are monotone. If $i \in [T + 1, T_1]$, we have

$$R_{1,i} - R_{1,i-1} = r_{1,i} = \mathbb{I}[x_i = \hat{y}_i] - \mathbb{I}[x_i = \hat{y}_{i-1}] = \mathbb{I}[x_i = \hat{y}_i] \geq 0,$$
$$S_{1,i} - S_{1,i-1} = s_{1,i} = \mathbb{I}[y_i = \hat{x}_i] - \mathbb{I}[y_i = \hat{x}_{i-1}] = \mathbb{I}[y_i = \hat{x}_i] \geq 0.$$

Therefore, $R_{1,T_1} \geq R_{1,T} \geq 0$ so after the first phase, both $R_{1,T_1}$ and $S_{1,T_1}$ are non-negative.

If $i \in [T_1 + 1, T_2]$ where $T_2 = T_1 + 4u + 50$, we have

$$R_{1,i} - R_{1,i-1} = r_{1,i} = \mathbb{I}[x_i = \hat{y}_i] - \mathbb{I}[x_i = \hat{y}_{i-1}] \geq \mathbb{I}[x_i = y_i] - 1 = 0,$$
$$S_{1,i} - S_{1,i-1} = s_{1,i} = \mathbb{I}[y_i = \hat{x}_i] - \mathbb{I}[y_i = \hat{x}_{i-1}] \geq \mathbb{I}[y_i = x_i] - 1 = 0.$$

Now we prove that $T_1 \leq T + 4c_0 + 50$. To prove this, we only need to show that $S_{1,i+4} - S_{1,i} \geq 1$ if $T \leq i \leq T_1 - 4$, so $S_{1,T+4c_0} \geq c_0 + S_{1,T} \geq 0$ if $T + 4c_0 \leq T_1 - 4$ and thus, $T_1 \leq T + 4c_0$. Actually, we have

$$S_{1,i+4} - S_{1,i} = s_{1,i+1} + s_{1,i+2} + s_{1,i+3} + s_{1,i+4}$$
$$= \mathbb{I}[y_{i+1} = \hat{x}_{i+1}] - \mathbb{I}[y_{i+1} = \hat{x}_i] + \mathbb{I}[y_{i+2} = \hat{x}_{i+2}] - \mathbb{I}[y_{i+2} = \hat{x}_{i+1}]$$
$$+ \mathbb{I}[y_{i+3} = \hat{x}_{i+3}] - \mathbb{I}[y_{i+3} = \hat{x}_{i+2}] + \mathbb{I}[y_{i+4} = \hat{x}_{i+4}] - \mathbb{I}[y_{i+4} = \hat{x}_{i+3}]$$
$$= \mathbb{I}[1 - \hat{y}_{i+1} = \hat{x}_{i+1}] + \mathbb{I}[1 - \hat{x}_{i+1} = 1 - \hat{y}_{i+1}] + \mathbb{I}[\hat{y}_{i+1} = 1 - \hat{x}_{i+1}] + \mathbb{I}[\hat{x}_{i+1} = \hat{y}_{i+1}]$$
$$= 2,$$

which is what we want to prove. An example of the first phase is shown as table 1.

Next, we prove that after the second phase, $\mathbb{E}_{T_2}^{1,1}(u) = 1$. Actually, we have for any $i \in [T_1 + 2, T_2]$,

$$R_{1,i} - R_{1,i-1} = r_{1,i} = \mathbb{I}[x_i = \hat{y}_i] - \mathbb{I}[x_i = \hat{y}_{i-1}] = \mathbb{I}[x_i = y_i] - \mathbb{I}[x_i = x_{i-1}] = 1,$$
$$S_{1,i} - S_{1,i-1} = s_{1,i} = \mathbb{I}[y_i = \hat{x}_i] - \mathbb{I}[y_i = \hat{x}_{i-1}] = \mathbb{I}[y_i = x_i] - \mathbb{I}[y_i = x_{i-1}] = 1.$$

Therefore, $R_{1,T_2} \geq R_{1,T_1+1} + 4u + 49 \geq R_{1,T_1} + 4u + 49 > u$ and $S_{1,T_2} \geq S_{1,T_1+1} + 4u + 49 \geq S_{1,T_1} + 4u + 49 > u$. Therefore, $\mathbb{E}_{T_2}^{1,1}(u) = 1$. An example of the first phase is shown as table 2.

| $\hat{x}_T = 0$ | $\hat{x}_{T+1} = 0$ | $\hat{x}_{T+2} = 1$ | $\hat{x}_{T+3} = 1$ | $\hat{x}_{T+4} = 0$ | $\hat{x}_{T+5} = 0$ | $\hat{x}_{T+6} = 1$ |
|---|---|---|---|---|---|---|
| $x_T$ | $x_{T+1} = 0$ | $x_{T+2} = 1$ | $x_{T+3} = 1$ | $x_{T+4} = 0$ | $x_{T+5} = 0$ | $x_{T+6} = 1$ |
| $y_T$ | $y_{T+1} = 1$ | $y_{T+2} = 1$ | $y_{T+3} = 0$ | $y_{T+4} = 0$ | $y_{T+5} = 1$ | $y_{T+6} = 1$ |
| $\hat{y}_T = 1$ | $\hat{y}_{T+1} = 0$ | $\hat{y}_{T+2} = 0$ | $\hat{y}_{T+3} = 1$ | $\hat{y}_{T+4} = 1$ | $\hat{y}_{T+5} = 0$ | $\hat{y}_{T+6} = 0$ |

Table 1: An example of Phase 1 for $\mathcal{P}_T$ with $T_1 = T + 6$.

| $\hat{x}_{T+6} = 1$ | $\hat{x}_{T+7} = 0$ | $\hat{x}_{T+8} = 1$ | $\hat{x}_{T+9} = 0$ | $\hat{x}_{T+10} = 1$ | $\hat{x}_{T+11} = 0$ | $\hat{x}_{T+12} = 1$ |
|---|---|---|---|---|---|---|
| $x_{T+6} = 1$ | $x_{T+7} = 0$ | $x_{T+8} = 1$ | $x_{T+9} = 0$ | $x_{T+10} = 1$ | $x_{T+11} = 0$ | $x_{T+12} = 1$ |
| $y_{T+6} = 1$ | $y_{T+7} = 0$ | $y_{T+8} = 1$ | $y_{T+9} = 0$ | $y_{T+10} = 1$ | $y_{T+11} = 0$ | $y_{T+12} = 1$ |
| $\hat{y}_{T+6} = 0$ | $\hat{y}_{T+7} = 0$ | $\hat{y}_{T+8} = 1$ | $\hat{y}_{T+9} = 0$ | $\hat{y}_{T+10} = 1$ | $\hat{y}_{T+11} = 0$ | $\hat{y}_{T+12} = 1$ |

Table 2: An example of Phase 2 for $\mathcal{P}_T$ with $T_1 = T + 6$ and $T_2 = T + 12$.

Until now, we have proved that $\mathcal{P}_T$ can lead to $\mathbb{E}_{T_2}^{1,1}(u) = 1$ where $T_2 \leq T + 4(c_0 + u) + 100$.

Then we lower-bound the probability of $\mathcal{P}_T$ happens given $\mathcal{H}_T$ such that $R_{1,T} \geq 0, S_{1,T} \geq -c_0$. Roughly speaking, the probability of each round in $\mathcal{P}$ is lower-bounded by a constant and we know that $\mathcal{P}_T$ has a limited number of rounds, which implies the entire probability of $\mathcal{P}_T$ is bounded by a power of the constant probability lower-bounding a single round in $\mathcal{P}_T$.

For a round $i$ in the first phase, because signals are i.i.d and $R_{1,i} \geq 0, S_{2,i} \geq 0$, the probability of $\{x_i = \hat{y}_{i-1}\} \wedge \{y_i = \hat{x}_{i-1}\} \wedge \{$Alice chooses $\mathrm{opt}_1\} \wedge \{$Bob chooses $\mathrm{opt}_2\}$ given a consistent history $\mathcal{H}_{i-1}$ is no less than $\min_{i,j \in \{0,1\}} \{P_{X,Y}(i,j)\} \times 0.25^2$. For a round $i$ in the second phase, because signals are i.i.d and $R_{1,i} \geq 0, S_{1,i} \geq 0$, the probability of $\{x_i = -x_{i-1}\} \wedge \{y_i = -x_{i-1}\} \wedge \{$Alice chooses $\mathrm{opt}_1\} \wedge \{$Bob chooses $\mathrm{opt}_1\}$ given a consistent history $\mathcal{H}_{i-1}$ is also no less than $\min_{i,j \in \{0,1\}} \{P_{X,Y}(i,j)\} \times 0.25^2$.

Therefore, for a history $\mathcal{H}_T$ such that $R_{1,T} \geq 0, S_{1,T} \geq -c_0$, we have

$$\Pr\left\{\left(\vee_{i=T}^{T+4(u+c_0)+100} \mathcal{E}_t^{1,1}(u)\right) \vee \left(\vee_{i=T}^{T+4(u+c_0)+100} \mathcal{E}_t^{2,2}(u)\right) = 1 \middle| \mathcal{H}_T\right\}$$
$$\geq \Pr\left\{\mathcal{P}_T \text{ happens} | \mathcal{H}_T\right\}$$
$$\geq \prod_{t=1}^{T_2} \left(\min_{i,j \in \{0,1\}} \{P_{X,Y}(i,j)\} \times 0.25^2\right)$$
$$\geq \left(\min_{i,j \in \{0,1\}} \{P_{X,Y}(i,j)\} \times 0.25^2\right)^{100+4(u+c_0)}.$$

Therefore, we can set $\lambda_u$ as $(\min_{i,j \in \{0,1\}} \{P_{X,Y}(i,j)\} \times 0.25^2)^{100+4(u+c_0)}$, which is what we want to prove. $\square$

### F.2.6   Proof of Lemma 4.9

*Proof.* By the definition of $c_0$, we know that when $\mathcal{E}_t^{1,1}(\lfloor \frac{u}{2} \rfloor) \vee \mathcal{E}_t^{2,2}(\lfloor \frac{u}{2} \rfloor) = 1$ happens, Alice and Bob will choose $\mathrm{opt}_1$ both with probability larger than $1 - \delta$ in the next $\zeta + 1$ rounds. Using this, we want to prove that $\mathbb{E}[\sum_{j=1}^{\zeta+1}(r_{1,t+j} - r_{2,t+j}) | \mathcal{H}_t$ such that $\mathcal{E}_t^{1,1}(\lfloor \frac{u}{2} \rfloor) = 1] \geq 100$. Actually, we can

deduce that

$$\mathbb{E}\left[\sum_{j=1}^{\zeta+1}(r_{1,t+j}-r_{2,t+j})\Bigg|\mathcal{H}_t \text{ such that } \mathcal{E}_t^{1,1}\left(\lfloor\frac{u}{2}\rfloor\right)=1\right]$$

$$=\sum_{j=1}^{\zeta+1}(\mathbb{E}\left[\mathbb{I}[x_{t+j}=\hat{y}_{t+j}]-\mathbb{I}[1-x_{t+j}=\hat{y}_{t+j}]\Big|\mathcal{H}_t \text{ such that } \mathcal{E}_t^{1,1}\left(\lfloor\frac{u}{2}\rfloor\right)=1\right]$$

$$-\mathbb{E}\left[\mathbb{I}[x_{t+j}=\hat{y}_{t+j-1}]-\mathbb{I}[1-x_{t+j}=\hat{y}_{t+j-1}]\Big|\mathcal{H}_t \text{ such that } \mathcal{E}_t^{1,1}\left(\lfloor\frac{u}{2}\rfloor\right)=1\right])$$

$$\geq -2-\sum_{j=2}^{\zeta+1}(\delta-(1-\delta)\gamma_1+\max\{\gamma_2,(1-\delta)\gamma_2-\delta\}) \qquad (16)$$

$$\geq -2+\sum_{j=2}^{\zeta+1}\frac{\gamma_1-\gamma_2}{2} \qquad (17)$$

$$> 100.$$

Here, eq. (16) is because when $j \geq 2$, we have

$$\sum_{j=1}^{\zeta+1}(\mathbb{E}\left[\mathbb{I}[x_{t+j}=\hat{y}_{t+j}]-\mathbb{I}[1-x_{t+j}=\hat{y}_{t+j}]\Big|\mathcal{H}_t \text{ such that } \mathcal{E}_t^{1,1}\left(\lfloor\frac{u}{2}\rfloor\right)=1\right]$$

$$-\mathbb{E}\left[\mathbb{I}[x_{t+j}=\hat{y}_{t+j-1}]-\mathbb{I}[1-x_{t+j}=\hat{y}_{t+j-1}]\Big|\mathcal{H}_t \text{ such that } \mathcal{E}_t^{1,1}\left(\lfloor\frac{u}{2}\rfloor\right)=1\right])$$

$$\geq -(1-\Pr\{\text{opt}_{t+j}^Y=\text{opt}_1\})+\Pr\{\text{opt}_{t+j}^Y=\text{opt}_1\}\gamma_1$$

$$+(1-\Pr\{\text{opt}_{t+j-1}^Y=\text{opt}_1\})-\Pr\{\text{opt}_{t+j-1}^Y=\text{opt}_1\}\gamma_2$$

$$> -\delta+(1-\delta)\gamma_1-\max\{\gamma_2,(1-\delta)\gamma_2-\delta\}.$$

Moreover, eq. (17) holds according to definition of $\delta$ in lemma 4.5. $\qquad\square$

### F.2.7  Proof of Lemma 4.10

*Proof.* Without loss of generality, we suppose $\mathcal{H}_T$ satisfies $\mathcal{E}_T^{1,1}(u)$, and what we need is to prove that $\exists u \in \mathbb{N}^+$ such that

$$\Pr\left\{\forall i \in \mathbb{N}, \mathcal{E}_{T+i(\zeta+1)}^{1,1}\left(\lfloor\frac{u}{2}\rfloor+i\right)=1\Big|\mathcal{H}_T\right\} \geq 1-\varepsilon,$$

for any $\varepsilon > 0$.

We choose $u$ to be larger than $2c_0+1$ at first. Then we let $\tau = \min_{t=T+i(\zeta+1),i\in\mathbb{N}^+}\{R_{1,t} \leq \lfloor\frac{u}{2}\rfloor \text{ or } S_{1,t} \leq \lfloor\frac{u}{2}\rfloor\}$ and $i_\tau = \lceil\frac{\tau-T}{\zeta+1}\rceil$. We construct a sequence of random variables $\{D_i\}_{i\in\mathbb{N}}$ such that

$$D_i = \left(R_{1,T+i(\zeta+1)}-R_{2,T+i(\zeta+1)}-100i\right)\mathbb{I}\left[T+(i-1)(\zeta+1)<\tau\right]$$

$$+\left(R_{1,T+i_\tau(\zeta+1)}-R_{2,T+i_\tau(\zeta+1)}-100i_\tau\right)\mathbb{I}\left[T+(i-1)(\zeta+1)\geq\tau\right].$$

Now we show that $\{D_i\}_{i=0,1,2,\cdots}$ is a sub-martingale with bounded difference, and we will use Azuma-Hoeffding inequality (theorem A.3) to show the value of $D_i$ (and thus $R_{1,t}$) cannot be too small. This implies that $\{D_j < u+2j-100j\}$ happens with a probability upper-bounded by a function of $u$. Therefore, the probability of $\{\forall i \in \mathbb{N}, \mathcal{E}_{T+i(\zeta+1)}^{1,1}\left(\lfloor\frac{u}{2}\rfloor+i\right)=1|\mathcal{H}_T\}$ can be lower-bounded by a decreasing function of $u$, which tends towards 1 when $u$ tends towards infinity.

Actually, we firstly verify that $\{D_i\}_{i\in\mathbb{N}}$ is a sub-martingale. If $\mathcal{H}_{T+(i-1)(\zeta+1)}$ satisfies that $\mathbb{I}\left[T+(i-1)(\zeta+1)<\tau\right]=1$, we have $R_{1,T+(i-1)(\zeta+1)}>\lfloor\frac{u}{2}\rfloor$ and $S_{1,T+(i-1)(\zeta+1)}>\lfloor\frac{u}{2}\rfloor$.

According to lemma 4.9, we have

$$
\mathbb{E}\left[D_i\big|\mathcal{H}_{T+(i-1)(\zeta+1)} \text{ with } D_{i-1}, \cdots, D_0\right]
$$
$$
= \mathbb{E}\left[\sum_{j=1}^{\zeta+1}\left(r_{1,T+(i-1)(\zeta+1)+j} - r_{2,T+(i-1)(\zeta+1)+j}\right)\bigg|\mathcal{H}_{T+(i-1)(\zeta+1)}\right]
$$
$$
+ D_{i-1} - 100
$$
$$
> D_{i-1}.
$$

If $\mathbb{I}\left[T + (i-1)(\zeta+1) < \tau\right] = 0$, we know that $D_i = D_{i-1}$.

When $i \leq i_\tau$, we know that $|D_i - D_{i-1}| = |2R_{1,T+i(\zeta+1)} - 2R_{1,T+(i-1)(\zeta+1)}| + 100 \leq 102 + \zeta$.
When $i > i_\tau$, $|D_i - D_{i-1}| = 0$.

Now according to Azuma-Hoeffding inequality (theorem A.3), we have

$$
\Pr\left\{D_j < u + 2j - 100j\right\} \leq \Pr\left\{D_j - D_0 < -98j - u\right\}
$$
$$
\leq \exp\left(\frac{-(98j+u)^2}{2j\left(102+\zeta\right)^2}\right).
$$

If $\wedge_{i=0}^{j-1}\mathcal{E}_{T+i(\zeta+1)}^{1,1}\left(\lfloor\frac{u}{2}\rfloor + i\right) = 1$ and $R_{1,T+j(\zeta+1)} \leq \lfloor\frac{u}{2}\rfloor + j\rfloor$, we can deduce that $i_\tau > j$ and hence,
$D_j < u + 2j - 100j$. We have

$$
\Pr\left\{\wedge_{i=0}^{j-1}\mathcal{E}_{T+i(\zeta+1)}^{1,1}\left(\lfloor\frac{u}{2}\rfloor + i\right) = 1, R_{1,T+j(\zeta+1)} \leq \lfloor\frac{u}{2}\rfloor + j\bigg|\mathcal{H}_T\right\}
$$
$$
\leq \Pr\{D_j < u + 2j - 100j\}
$$
$$
\leq \exp\left(\frac{-(98j+u)^2}{2j\left(102+\zeta\right)^2}\right).
$$

Symmetrically, we have

$$
\Pr\left\{\wedge_{i=0}^{j-1}\mathcal{E}_{T+i(\zeta+1)}^{1,1}\left(\lfloor\frac{u}{2}\rfloor + i\right) = 1, S_{1,T+j(\zeta+1)} \leq \lfloor\frac{u}{2}\rfloor + j\bigg|\mathcal{H}_T\right\}
$$
$$
\leq \exp\left(\frac{-(98j+u)^2}{2j\left(102+\zeta\right)^2}\right).
$$

Therefore, we have

$$
\Pr\left\{\wedge_{i=0}^{j-1}\mathcal{E}_{T+i(\zeta+1)}^{1,1}\left(\lfloor\frac{u}{2}\rfloor + i\right) = 1, \mathcal{E}_{T+j(\zeta+1)}^{1,1}\left(\lfloor\frac{u}{2}\rfloor + j\right) = 0\bigg|\mathcal{H}_T\right\}
$$
$$
\leq \Pr\left\{\wedge_{i=0}^{j-1}\mathcal{E}_{T+i(\zeta+1)}^{1,1}\left(\lfloor\frac{u}{2}\rfloor + i\right) = 1, R_{1,T+j(\zeta+1)} \leq \lfloor\frac{u}{2}\rfloor + j\bigg|\mathcal{H}_T\right\}
$$
$$
+ \Pr\left\{\wedge_{i=0}^{j-1}\mathcal{E}_{T+i(\zeta+1)}^{1,1}\left(\lfloor\frac{u}{2}\rfloor + i\right) = 1, S_{1,T+j(\zeta+1)} \leq \lfloor\frac{u}{2}\rfloor + j\bigg|\mathcal{H}_T\right\}
$$
$$
\leq 2\exp\left(\frac{-(98j+u)^2}{2j\left(\zeta+102\right)^2}\right).
$$

Furthermore, we can bound the probability of $\Pr\{\forall i \in \mathbb{N}, \mathcal{E}^{1,1}_{T+(\zeta+1)i}\left(\lfloor\frac{u}{2}\rfloor + i\right) \vee \mathcal{E}^{2,2}_{T+(\zeta+1)i}\left(\lfloor\frac{u}{2}\rfloor + i\right) = 1 | \mathcal{H}_T\}$ as

$$\Pr\left\{\forall i \in \mathbb{N}, \mathcal{E}^{1,1}_{T+(\zeta+1)i}\left(\lfloor\frac{u}{2}\rfloor + i\right) = 1 \Big| \mathcal{H}_T\right\}$$

$$\geq 1 - \sum_{j=1}^{+\infty} \Pr\left\{\wedge_{i=0}^{j-1}\mathcal{E}^{1,1}_{T+i(\zeta+1)}\left(\lfloor\frac{u}{2}\rfloor + i\right) = 1, \mathcal{E}^{1,1}_{T+j(\zeta+1)}(\lfloor\frac{u}{2}\rfloor + j) = 0\Big|\mathcal{H}_T\right\}$$

$$\geq 1 - \sum_{j=1}^{+\infty} 2\exp\left(\frac{-(98j+u)^2}{2j\left(\zeta+102\right)^2}\right)$$

$$\geq 1 - \sum_{j=1}^{+\infty} 2\exp\left(\frac{-4802j - 98u}{\left(\zeta+102\right)^2}\right)$$

$$= 1 - 2\exp\left(\frac{-98u}{\left(\zeta+102\right)^2}\right)\exp\left(\frac{-4802}{\left(\zeta+102\right)^2}\right)\frac{1}{1 - \exp\left(\frac{-j}{(\zeta+102)^2}\right)}$$

$$= 1 - \frac{2\exp\left(\frac{-98u+4802+j}{(\zeta+102)^2}\right)}{\exp\left(\frac{j}{(\zeta+102)^2}\right) - 1},$$

where $\frac{2\exp\left(\frac{-98u+4802+j}{(\zeta+102)^2}\right)}{\exp\left(\frac{j}{(\zeta+102)^2}\right) - 1}$ decays exponentially in $u$. Until now, we can find a sufficient large $u$ such that $\Pr\{\forall i \in \mathbb{N}, \mathcal{E}^{1,1}_{T+(\zeta+1)i}\left(\lfloor\frac{u}{2}\rfloor + i\right) \vee \mathcal{E}^{2,2}_{T+(\zeta+1)i}\left(\lfloor\frac{u}{2}\rfloor + i\right) = 1|\mathcal{H}_T\} \geq 1 - \varepsilon$ for $\mathcal{H}_T$ with $\mathcal{E}^{1,1}_T(u) \wedge \mathcal{E}^{2,2}_T(u) = 1$. $\qquad\square$

### F.2.8 Proof of Truthful Convergence (Theorem 4.1)

Before the proof of our final theorem, we introduce a lemma showing that good events happen for infinite many times with probability 1. Combining lemma 4.7 in step 2 and lemma 4.8, we use the martingale theory to get the following lemma.

**Lemma F.1.** *Given the game defined in theorem 4.1, for all $u$ we have $\Pr\{\limsup_{t\to+\infty} \mathcal{E}^{1,1}_t(u) \vee \mathcal{E}^{2,2}_t(u) = 1\} = 1$.*

*Proof.* Initially, we prove that in order to prove $\Pr\{\limsup_{t\to+\infty} \mathcal{E}^{1,1}_t(u) \vee \mathcal{E}^{2,2}_t(u) = 1\} = 1$, we only need to prove

$$\Pr\{\forall t \in \mathbb{N}^+, \mathcal{E}^{1,1}_t(u) \vee \mathcal{E}^{2,2}_t(u) = 0\} = 0.$$

Actually, in order to prove $\Pr\{\limsup_{t\to+\infty} \mathcal{E}^{1,1}_t(u) \vee \mathcal{E}^{2,2}_t(u) = 1\} = 1$, we only need to prove that for any $u \in \mathbb{N}^+$, $\Pr\{\exists t \in \mathbb{N}^+, \mathcal{E}^{1,1}_t(u) \vee \mathcal{E}^{2,2}_t(u) = 1\} = 1$. More specifically, if this claim holds, we can always find a $t_1 \in \mathbb{N}^+$ such that $\mathcal{E}^{1,1}_{t_1}(u_1) \vee \mathcal{E}^{2,2}_{t_1}(u_1) = 1$ where $u_1 > u$, a $t_2 > t_1$ such that $\mathcal{E}^{1,1}_{t_2}(u_2) \vee \mathcal{E}^{2,2}_{t_2}(u_2) = 1$ where $u_2 > u_1$, a $t_3 > t_2$ such that $\mathcal{E}^{1,1}_{t_3}(u_3) \vee \mathcal{E}^{2,2}_{t_3}(u_3) = 1$ where $u_3 > u_2$ and so on. Hence, we can find a sequence $t_1, t_2, \cdots$ such that $\mathcal{E}^{1,1}_{t_i}(u) \vee \mathcal{E}^{2,2}_{t_i}(u) = 1$ for every $i \in \mathbb{N}^+$, which indicates that $\Pr\{\limsup_{t\to+\infty} \mathcal{E}^{1,1}_t(u) \vee \mathcal{E}^{2,2}_t(u) = 1\} = 1$. In order to prove $\Pr\{\exists t \in \mathbb{N}^+, \mathcal{E}^{1,1}_t(u) \vee \mathcal{E}^{2,2}_t(u) = 1\} = 1$, we only need to show that $\Pr\{\forall t \in \mathbb{N}^+, \mathcal{E}^{1,1}_t(u) \vee \mathcal{E}^{2,2}_t(u) = 0\} = 0$.

We know from lemma 4.7 that with probability 1, there are infinitely many $t \in \mathbb{N}^+$ such that $\overline{\mathcal{E}^{1,2}_t} \vee \mathcal{E}^{2,1}_t = 1$. For any $\mathcal{H}_t$ such that $\overline{\mathcal{E}^{1,2}_t \vee \mathcal{E}^{2,1}_t} = 1$, we can find a $\{\mathcal{E}^{1,1}_i(u) \vee \mathcal{E}^{2,2}_i(u) = 1\}$ in the next $100 + 4(c_0 + u)$ rounds with probability no less than $\lambda_u$ according to Lemma 4.8. Therefore,

we can create a sub-martingale

$$D_i = \{\sum_{j=0}^{i} \sum_{t=T_j}^{T_j+100+4(c_0+u)} \mathcal{E}_t^{1,1}(u) \vee \mathcal{E}_t^{2,2}(u) \text{ where } T_j \text{ is the smallest number such that}$$

$$\sum_{t=1}^{T_j} \overline{\mathcal{E}_t^{1,2} \vee \mathcal{E}_t^{2,1}} = (4c_0 + 4u + 101)j\} - \lambda_u i.$$

Now we show that $\{D_i\}_{i=0,1,2,\cdots}$ is indeed a sub-martingale with bounded difference, and we will use Azuma-Hoeffding inequality (theorem A.3) to show the value of $D_i$ cannot be too big. Therefore, $D_i \leq k - \lambda_u i$ for any $k \in \mathbb{N}^+$ such that $k < i\lambda u$ can only happen finitely many times by Borel-Cantelli lemma (theorem A.6). This implies $\limsup_{t\to+\infty} \mathcal{E}_t^{1,1}(u) \vee \mathcal{E}_t^{2,2}(u)$ happens with probability 1.

This is because

$$\mathbb{E}[D_i|\mathcal{H}_{T_i}] \geq -\lambda_u i + \sum_{j=0}^{i-1} \sum_{t=T_j}^{T_j+100+4(c_0+u)} \mathcal{E}_t^{1,1}(u) \vee \mathcal{E}_t^{2,2}(u)$$

$$+ \mathbb{E}\left[\sum_{t=T_i}^{T_i+100+4(c_0+u)} \mathcal{E}_t^{1,1}(u) \vee \mathcal{E}_t^{2,2}(u) \middle| \overline{\mathcal{E}_{T_i}^{1,2} \vee \mathcal{E}_{T_i}^{2,1}} = 1\right]$$

$$\geq D_{i-1} + \lambda_u(i-1) + \lambda_u - \lambda_u i \geq D_{i-1}.$$

Moreover, we know that $|D_{i+1} - D_i| \leq 4c_0 + 4u + 101$ for $i \in \mathbb{N}^+$, therefore, according to Azuma-Hoeffding inequality (theorem A.3), we have

$$\Pr\{D_i \leq k - \lambda_u i\} \leq \Pr\{D_i - D_0 \leq k - \lambda_u i\}$$

$$\leq \exp\left(\frac{-(k - \lambda_u i)}{2i(101 + 4u + 4c_0)^2}\right),$$

which decays exponentially in $i$. This holds for any $k \in \mathbb{N}^+$ and $i > \frac{k}{\lambda_u}$. Therefore, by Borel-Cantelli lemma (theorem A.6), $\{D_i \leq k - \lambda_u i\}$ will happen only finitely often almost surely. If $\{D_i \leq k - \lambda_u i\}$ does not happen, we can deduce that $\mathcal{E}_t^{1,1}(u) \vee \mathcal{E}_t^{2,2}(u)$ happens for at least $k$ times in the first $T_i + 4c_0 + 4u + 100$ rounds. Hence, $\mathcal{E}_t^{1,1}(u) \vee \mathcal{E}_t^{2,2}(u)$ happens with probability 1 and consequentially, $\Pr\{\limsup_{t\to+\infty} \mathcal{E}_t^{1,1}(u) \vee \mathcal{E}_t^{2,2}(u) = 1\} = 1$. □

Now combining lemma F.1 and lemma 4.10, we complete our final proof (theorem 4.1).

*Proof.* First note that if $\mathcal{E}_T^{\text{Convergence Condition}} = \{\forall i \in \mathbb{N}, \mathcal{E}_{T+\left(\lceil\frac{1000}{\gamma_1-\gamma_2}\rceil+1\right)i}^{1,1}\left(\lfloor\frac{u}{2}\rfloor + i\right) \vee$

$\mathcal{E}_{T+\left(\lceil\frac{1000}{\gamma_1-\gamma_2}\rceil+1\right)i}^{2,2}\left(\lfloor\frac{u}{2}\rfloor + i\right) = 1\}$ happens for some $T$, we can deduce that $\lim_{t\to+\infty}(R_{1,t}-R_{2,t}) = \lim_{t\to+\infty}(S_{1,t} - S_{2,t}) = \pm\infty$, and $\Pr\{\lim_{t\to+\infty}\text{opt}_t^X = \lim_{t\to+\infty}\text{opt}_t^Y = \text{opt}_1\} = 1$ or $\Pr\{\lim_{t\to+\infty}\text{opt}_t^X = \lim_{t\to+\infty}\text{opt}_t^Y = \text{opt}_2\} = 1$. Therefore, $\Pr\{\vee_{T=1}^{+\infty}\mathcal{E}_T^{\text{Convergence Condition}}\} = 1$ implies theorem 4.1.

Otherwise, suppose $\Pr\{\vee_{T=1}^{+\infty}\mathcal{E}_T^{\text{Convergence Condition}} = 1\} = 1 - \varepsilon_2 < 1$. Let $\varepsilon_1 = \frac{1}{2}\varepsilon_2$ and $u_1$ satisfy lemma 4.10, then we can deduce that

$$\Pr\left\{\vee_{T=1}^{+\infty}\mathcal{E}_T^{\text{Convergence Condition}}\right\}$$

$$\geq \sum_{T=1}^{+\infty}\Pr\left\{\vee_{t=1}^{+\infty}\mathcal{E}_t^{\text{Convergence Condition}}, \vee_{t=1}^{T-1}\mathcal{E}_t^{1,1}(u_1) \vee \mathcal{E}_t^{2,2}(u_1) = 0, \mathcal{E}_T^{1,1}(u_1) \vee \mathcal{E}_T^{2,2}(u_1) = 1\right\}$$

$$= \sum_{T=1}^{+\infty}\Pr\left\{\vee_{t=1}^{+\infty}\mathcal{E}_t^{\text{Convergence Condition}}\,\bigg|\,\left(\vee_{t=1}^{T-1}\mathcal{E}_t^{1,1}(u_1) \vee \mathcal{E}_t^{2,2}(u_1) = 0\right) \wedge \left(\mathcal{E}_T^{1,1}(u_1) \vee \mathcal{E}_T^{2,2}(u_1) = 1\right)\right\}$$

$$\times \Pr\left\{\vee_{t=1}^{T-1}\mathcal{E}_t^{1,1}(u_1) \vee \mathcal{E}_t^{2,2}(u_1) = 0, \mathcal{E}_T^{1,1}(u_1) \vee \mathcal{E}_T^{2,2}(u_1) = 1\right\}$$

$$\geq \sum_{T=1}^{+\infty}\Pr\left\{\mathcal{E}_T^{\text{Convergence Condition}}\,\bigg|\,\left(\vee_{t=1}^{T-1}\mathcal{E}_t^{1,1}(u_1) \vee \mathcal{E}_t^{2,2}(u_1) = 0\right) \wedge \left(\mathcal{E}_T^{1,1}(u_1) \vee \mathcal{E}_T^{2,2}(u_1) = 1\right)\right\}$$

$$\times \Pr\left\{\vee_{t=1}^{T-1}\mathcal{E}_t^{1,1}(u_1) \vee \mathcal{E}_t^{2,2}(u_1) = 0, \mathcal{E}_T^{1,1}(u_1) \vee \mathcal{E}_T^{2,2}(u_1) = 1\right\}$$

$$\geq \sum_{T=1}^{+\infty}\min_{T'\in\mathbb{N}^+, \mathcal{H}_T'\in\mathcal{E}_{T'}^{1,1}(u_1)\vee\mathcal{E}_{T'}^{2,2}(u_1)}\Pr\left\{\mathcal{E}_{T'}^{\text{Convergence Condition}}\,\bigg|\,\mathcal{H}_{T'}\right\}$$

$$\times \Pr\left\{\vee_{t=1}^{T-1}\mathcal{E}_t^{1,1}(u_1) \vee \mathcal{E}_t^{2,2}(u_1) = 0, \mathcal{E}_T^{1,1}(u_1) \vee \mathcal{E}_T^{2,2}(u_1) = 1\right\}$$

$$= \min_{T\in\mathbb{N}^+, \mathcal{H}_T\in\mathcal{E}_T^{1,1}(u_1)\vee\mathcal{E}_T^{2,2}(u_1)}\Pr\left\{\mathcal{E}_T^{\text{Convergence Condition}}\,\bigg|\,\mathcal{H}_T\right\}\Pr\left\{\vee_{t=1}^{+\infty}\mathcal{E}_t^{1,1}(u_1) \vee \mathcal{E}_t^{2,2}(u_1) = 1\right\}$$

$$= \min_{T\in\mathbb{N}^+, \mathcal{H}_T\in\mathcal{E}_T^{1,1}(u_1)\vee\mathcal{E}_T^{2,2}(u_1)}\Pr\left\{\mathcal{E}_T^{\text{Convergence Condition}}\,\bigg|\,\mathcal{H}_T\right\} \qquad \text{(by lemma F.1)}$$

$$\geq 1 - \varepsilon_1, \qquad\qquad\qquad\qquad\qquad\qquad\qquad\qquad\qquad\qquad \text{(by lemma 4.10)}$$

which is a contradiction. $\qquad\qquad\qquad\qquad\qquad\qquad\qquad\qquad\qquad\qquad\qquad\qquad\qquad\qquad$ $\square$

## G   Error Bars of Simulation in Section 5

We run our simulations single-threadedly on AMD Ryzen™ R7-5700U Processor at 3.60GHz with 16GB DDR4 SDRAM. The total running time of all the simulations in fig. 2 is $421$ seconds. Code is available in `https://github.com/fengtony686/peer-prediction-convergence`. We give error bars of converge rates of each algorithm shown in fig. 2 one by one. Each error bar is drawn by computing converge proportion of each algorithm in $400$ repeating simulations for $10$ times.

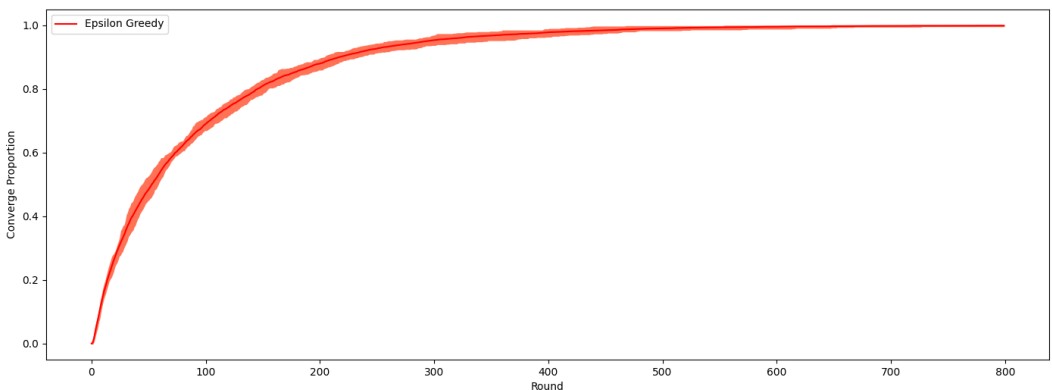

Figure 3: Error Bar of $\epsilon$-Greedy in Section 5.

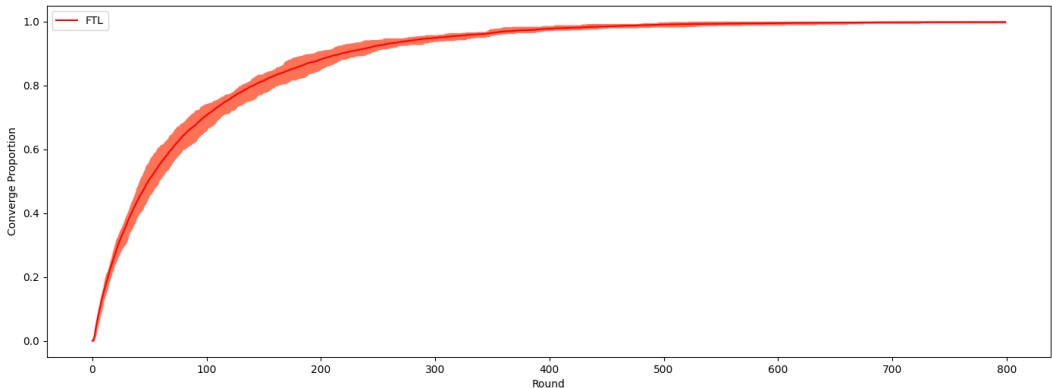

Figure 4: Error Bar of FTL in Section 5.

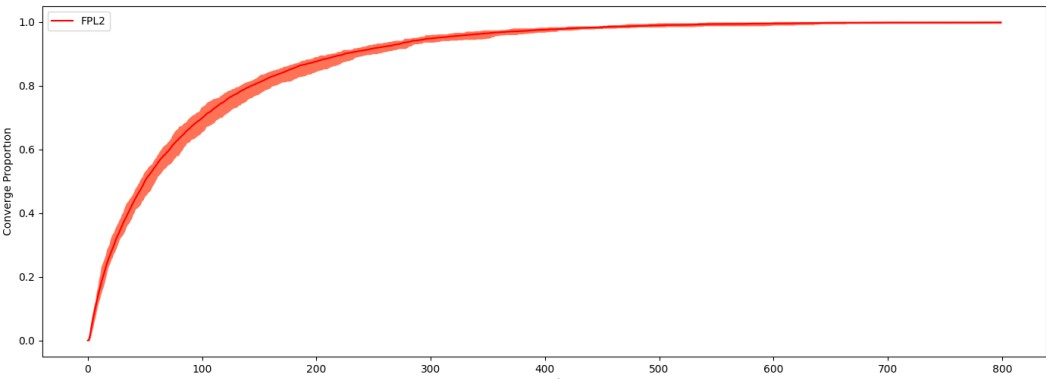

Figure 5: Error Bar of FPL2 in Section 5.

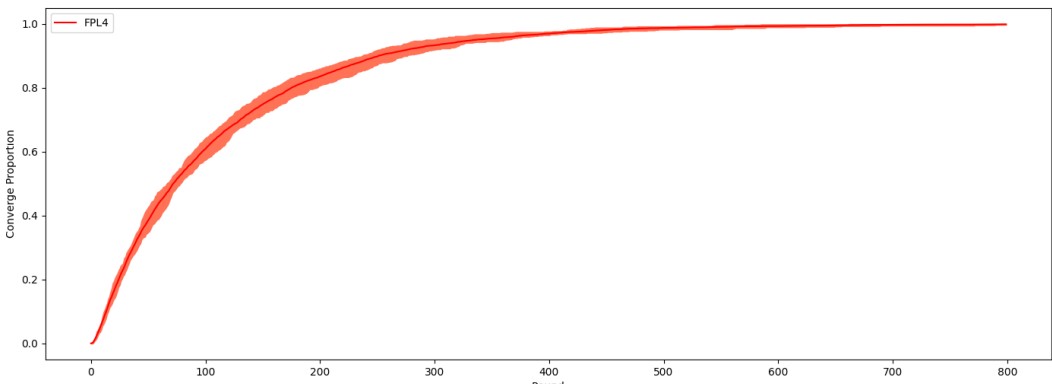

Figure 6: Error Bar of FPL4 in Section 5.

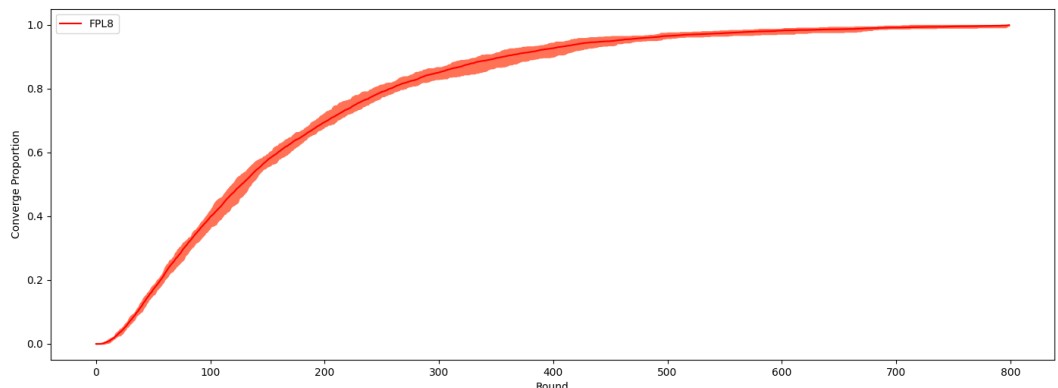

Figure 7: Error Bar of FPL8 in Section 5.

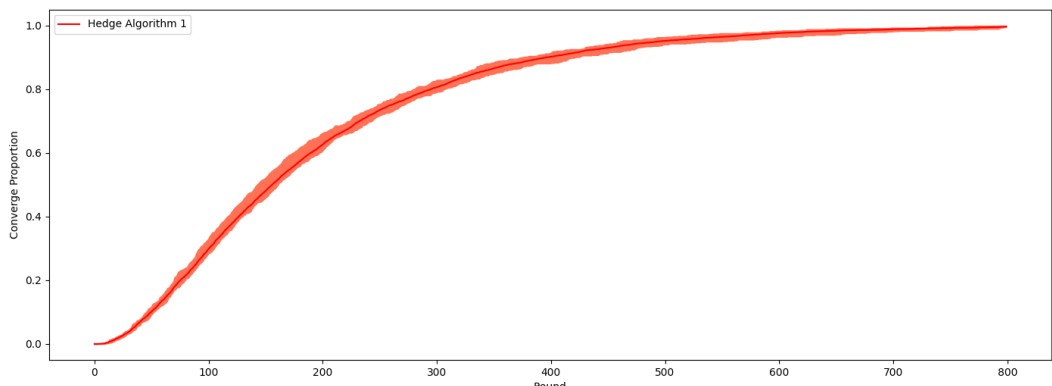

Figure 8: Error Bar of Hedge Algorithm 1 in Section 5.

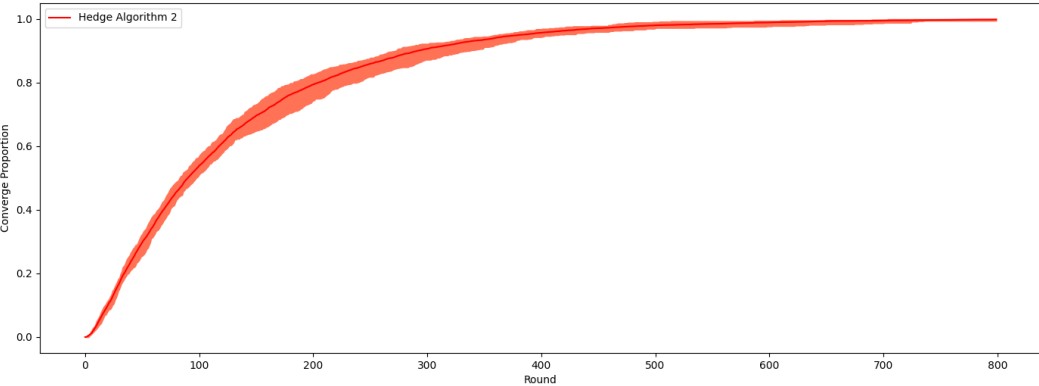

Figure 9: Error Bar of Hedge Algorithm 2 in Section 5.