# OpenReview forum: "Peer Prediction for Learning Agents"
_NeurIPS.cc/2022/Conference — NeurIPS 2022 Accept_

### Official Review · Reviewer_CRBk · 2022-07-03

**Rating:** 5
**Confidence:** 3
**Soundness:** 3 good
**Presentation:** 1 poor
**Contribution:** 3 good

**Summary:**

This paper studies a sequential reporting problem where two agents are incentivized via payments to report back a private signal each of them receives. While this problem has been studied under strong assumptions in the Bayesian setting, the authors are the first to study this problem when the two agents are learners and can update their actions to reflect what they have learned about the rewards based on this history so far. The author’s show that no-regret learning on its own is too weak a notion for any algorithm to incentivize truth-telling in the limit. But, the main punchline of the paper is that for a large class of natural learning algorithms based on reward updating (included in this class are several no-regret algorithms such as multiplicative weights and follow-the-leader), truthful peer prediction is indeed possible using a classic scheme (CA) that (to my understanding) incentivizes agreement between the two agents. This phenomenon is also demonstrated via an experiment.

**Questions:**

Line 97-98: State that X_t and Y_t denote random variables, and x_t and y_t are their realizations.

Section 2 would benefit from some more intuition and motivation. For example the “correlated agreement mechanism” is new to me, and a sentence or two on why it is important/intuitive in this setting would have been helpful.

Lines 116-118: the remark about multi-task information elicitation is confusing to someone who doesn’t already know what this is.

Lines 120-130: I found it difficult to understand the reasoning behind this setup. Why are the four actions defined as they are? If a strategy is a distribution over these four action that depends on the history of signals, isn’t it equivalent to let the set of actions be {0,1}, and then a strategy would be a mapping into the set of distributions on {0,1} that depends on the history/knowledge so far? I’m not able to see the need to have four distinguished options as they are defined currently, so if this is an important assumption the reason for it should be greatly clarified.

Line 152: “permutation equilibrium” (which I don’t think is a standard solution concept) has not been defined.

Lines 157-159: My understanding is that the goal is to design algorithms to incentivize truthful reporting. I roughly understand that if both agents converge to opt_2 (fully mis-reporting), then the algorithm can be “flipped” to revert back to truthful reporting. But I think this needs more elaboration. How exactly can this be done with a single bit of “whether the prior of 0 is larger than 1”? There hasn’t been any discussion of priors/beliefs at this point. Also, would this allow for Defn 2.3 to wlog only include opt_1? If so that might be a more intuitive way to write the condition.

In assumption 3.3 should it be “nondecreasing” rather than “nonincreasing”? An intuitive explanation in words of what f and the assumptions mean would be helpful, though the instantiation for Multplicative weights is helpful.

Line 222: say that FPL is follow the perturbed leader.

General stylistic comments: many grammatical errors throughout. Avoid forward references (e.g. line 163).


**Strengths And Weaknesses:**

Given that the CA scheme plays such an important role in the main results of the paper, I would have liked to see more discussion on (1) why it is important (2) what it means. It seems to reward one agent when the other agent changes her prediction to agree with that agent. But as far as I could tell there was no further discussion on this.

Also, more discussion on the class of reward-based learning algorithms would have been helpful, as this is the main condition posed by the authors as a fix to the unrealistic behavior that is possible under only a no-regret assumption. Why, intuitively, does this fix those issues? Can’t agents still collude in this model? To me it seems like the idea of reward updating a neat one, but its presentation is opaque: what do the axioms (assumptions 3.2-3.4) mean in plain language? Why use these specific axioms?

The analysis of the stochastic process induced by CA and the learning agents is an interesting technical contribution, though I did not read the proofs in detail.

I think the overall contribution is quite interesting and nicely fills in a gap in the peer-prediction literature (agents that learn is an important wrinkle being studied in various settings), but I think the main weakness of this paper is its presentation. I found concepts to be introduced in an ad-hoc manner without much discussion on how things connected (specific examples in the “Questions” section).

---

> ### Author Response · Authors · 2022-08-02
> **Response to Reviewer CRBk**
>
> Thank you very much for your detailed review.  We have incorporated many of your suggestions into our updated version.
>
> **Why CA**
>
> The CA mechanism is the first multi-task peer prediction mechanism, and the CA mechanism possesses a similar structure with several follow-up mechanisms (pairing mechanism, DMI mechanism). Intuitively, the CA mechanism rewards agreement on the same task and punishes agreement on un-correlated tasks. We’ve added the intuition after introducing the CA mechanism.
>
> **Why reward based algorithm**
>
> In section 2, we provide several properties of the family of reward-based algorithms and their connection to several popular online learning algorithms. Intuitively, since agents’ decisions can only depend on the accumulated rewards in a reward-based algorithm, they can’t coordinate on some other aspects of the game, such as the round numbers, which excludes the collusion that we identified in theorem 3.1.
>
> **Lines 116-118: the remark about multi-task information elicitation**
>
> We rewrite section 2 and now distinguish between multi-task and sequential information elicitation settings. Due to space limitations, we defer most of our discussion about multi-task information elicitation and rational agents to Appendix B.
>
> **Lines 120-130: Why are the four actions defined as they are?**
>
> This is a great question.  When considering a one-round game, in the language of game theory, our $\text{opt}_i$’s are pure strategies of agents. (In fact, we now revised our paper to call them pure strategies, to be consistent with the standard language.) A pure strategy is a deterministic mapping from an agent’s signal to her action. A mixed strategy is a probability distribution over the pure strategies. Notions of game-theoretic equilibrium are usually defined for pure and mixed strategies. When extending the one-round game to sequential learning settings, where agents can change their strategies from round to round, we think it’s natural to consider their sequence of mixed strategies where their choice of a mixed strategy in each round depends on the history. Previous work Shnayder et al. [26] also considered the sequence of agents’ mixed strategies in sequential settings.
>
> However, what the reviewer suggested is another interesting way of extending the one-round game setting to sequential learning settings. In standard games, another notion of strategies is behavior strategies, which map from an agent’s signals to a distribution over actions {0, 1}. For many games, mixed strategies and behavioral strategies lead to the same observed behavior. The reviewer’s suggestion is to have learning agents make decisions on what behavioral strategy to use in each round in the sequential setting.
>
> We find the mixed strategy setup to be arguably more natural because the most desirable outcome of the mechanism, from the designer’s perspective, is that all agents play the truthful reporting pure strategy $\text{opt}_1$.
>
> **How exactly can flipping be done with a single bit of “whether the prior of 0 is larger than 1”**
>
> The single bit of information refers to whether an agent (e.g. Alice) is more likely to have $0$ signal or $1$ signal across all tasks. If we know this information, we can use the empirical frequency of Alice’s reported signal to tell apart whether the mechanism has converged to both players playing $\text{opt}_1$ or both players playing $\text{opt}_2$, if the number of tasks is large enough.
>
> **Line 152: “permutation equilibrium” (which I don’t think is a standard solution concept) has not been defined.**
>
> Permutation equilibrium is used in the peer prediction literature. It refers to an equilibrium where agents’ strategy is a fixed permutation of their signals. In our setting, both agents playing $\text{opt}_2$ is a permutation equilibrium in CA. We have made this more explicit in our revised version of the paper.

---

### Official Review · Reviewer_ikCW · 2022-07-11

**Rating:** 5
**Confidence:** 1
**Soundness:** 3 good
**Presentation:** 3 good
**Contribution:** 3 good

**Summary:**

This paper aims to theoretically analyze the dynamics of sequential peer prediction mechanisms, especially when participants are learning agents. To achieve this goal the authors firstly analyze the limitation of the notion no regret, that is, even no regret algorithms may also not converge. Then, the authors put their focus on a specific type of algorithms, where the policy is updated only based on the agents' cumulative rewards. In the experiments, the authors have conducted many simulation studies to demonstrate the effectiveness of their model.

**Questions:**

See the Strengths And Weaknesses

**Limitations:**

Yes, the authors have discussed the limitations of their model.

**Strengths And Weaknesses:**


To be honest, the topic of this paper is too far from my research domains. I can only make evaluations based on my general knowledge. In general, I think the paper is very well written, the organization is comfortable and compact. The studied problem seems to be important. Extending traditional static information elicitation mechanisms to sequential settings is important and should be more practical in real-world problems. The theory seems to be sound, which can well support their claims.
My concerns are as follows: (1) it could be better if there are some real-world experiments. (2) If the sequence is too long, whether the error produced from the former steps influence much on the following steps.

As mentioned before, my background does not fit this paper, thus if other reviewers have negative opinions, I can change my score.

---

> ### Author Response · Authors · 2022-08-02
> **Response to Reviewer ikCW**
>
> Thank you very much for your review.  The second question about the influence of the error in the former steps is very interesting.  Though our results do not directly address this issue, the reward-based algorithms are intuitively robust against a few rounds with error.  Because the decision of the algorithms only depends on the accumulated reward, a few rounds with error cannot significantly change the accumulated reward and hence the convergence of reward-based algorithms.

---

### Official Review · Reviewer_sA5q · 2022-07-11

**Rating:** 6
**Confidence:** 4
**Soundness:** 4 excellent
**Presentation:** 3 good
**Contribution:** 3 good

**Summary:**

This paper considers the sequential peer prediction setting when the participants are learning agents. In particular, the authors ask the following question — if a peer prediction mechanism is used to reward the learning agents, then under what conditions they converge to truthful reporting. The first result shows that some no regret learning mechanisms might fail to converge to truthful reporting. This is because the agents can construct a no-regret sequence of reports irrespective of their private signals.

The main contribution lies in showing that convergence to truthful strategies is possible if imposes certain assumptions on the no-regret learning algorithms. Fortunately, this class includes standard no-regret learning algorithms like Hedge, FPL etc. In particular, the main result shows that if the designer uses correlated agreement mechanism, and the joint distribution over the signals satisfy positive correlation, then convergence to truthful strategy is possible.

**Questions:**

Some questions for the authors.
- Does the current proof work with bandit feedback e.g. what happens if one agent doesn't observe the other agent's reported signal and only receives the scores?
- In simulation, all the mechanisms eventually converge to truthful reporting. Is it possible to empirically show that there exists some mechanism which does not converge? I understand the existence of such a mechanism was shown using the probabilistic method, so it might not be possible to empirically show the existence of such a mechanism.

**Limitations:**

This paper doesn't have any negative societal impact.

**Strengths And Weaknesses:**


Strengths:
- I think it is an interesting problem to address the convergence of learning algorithms under peer prediction based payment mechanisms. Prior work has empirically studied convergence properties of learning algorithms, and to the best of my knowledge, this paper proves convergence to truthful strategy for the first time.
- I found the analysis of the convergence proof quite interesting and non-trivial. The authors also clearly explained different steps involved in the proof.

Weaknesses:
- The setting only considers binary signal setting and that certainly makes the analysis easier.
- My other criticism is regarding the setting. If the designer knows that the agents will use a learning algorithm then the designer doesn’t have to use CA mechanism every round. Since all the tasks are identically distributed the designer can eventually learn the prior distribution and might not even need peer prediction mechanism. Is this observation accurate or am I missing something here?
- Finally, the proof assumes full information feedback setting. This is because an agent (say Alice) can observe Bob’s signal and can compute exact payment under a different strategy. However, in practice e.g. in rating platforms, people just receive their score for a report. So the learning algorithms must run with bandit feedback. I think this is the main limitation of the current paper.

Some other comments:
- Section 2 introduces the concept of rank k mechanism. Is this definition used anywhere? I understand that CA is a rank-2 mechanism, I am not sure if k > 2 is used anywhere. If not, it might be a good idea to avoid this definition.

---

> ### Author Response · Authors · 2022-08-02
> **Response to Reviewer sA5q**
>
> Thank you very much for your time and effort in reviewing our paper. Please find our response below.
>
> **Extend to the case of non-binary signal/reports**
>
> During the simulation, we observed that when the signals are non binary (ternary), Alice and Bob using FPL may converge to nontruthful (and non-permutation) strategies in the CA mechanism.   This result indicates that using reward-based online learning algorithms is not sufficient for truthful convergence under the CA mechanism.
>
> Additionally, previous work by Shnayder et al. also shows similar nontruthful convergence holds when agents use continuous time learning dynamics when the signals are non-binary.
>
> We believe extending to nonbinary settings is an exciting future direction but better mechanisms or stronger assumptions than positively correlated assumptions are needed.
>
> **If the designer knows that the agents will use a learning algorithm**
>
> We want to emphasize that the ultimate goal of the designer is to elicit agents’ private signals. Take peer review as an example.  We need their reviews (private signals) on each paper instead of the distribution of their reviews. Many peer prediction mechanisms even assume that the designer knows the prior distribution of agent signals.
>
> Another reason for using the CA mechanism is that we’d like the mechanism to be robust for both rational agents and learning agents and CA is known to already provide good incentives for rational agents. The reviewer’s suggestion opens up a very interesting future direction, that is, to design mechanisms that have desirable properties for learning agents.
>
> **Does the current proof work with bandit feedback?**
>
> Note that reward-based algorithms cannot be applied to the bandit feedback setting, because cumulative rewards require full feedback.   Thus, our proof for truthful convergence on reward-based algorithms cannot be applied to the bandit feedback.
>
> We have observed that the UCB algorithm, a bandit feedback algorithm, cannot achieve truthful convergence under the CA mechanism.  Informally, UCB always explores all strategies infinitely number of times and cannot achieve truthful convergence. It will be interesting to design another family of learning algorithms in bandit feedback setting that can achieve truthful convergence.
>
> **Concept of rank $k$ mechanism**
>
> This definition is also used in theorem 3.1. Theorem 3.1 applies to all sequential information elicitation mechanisms with bounded rank $k$.
>
> **Is it possible to empirically show that there exists some mechanism which does not converge?**
>
> We indeed empirically observe that one no-regret algorithm UCB does not converge to any option. UCB is a bandit feedback algorithm. It remains interesting to understand whether some full-feedback algorithms do not achieve truthful convergence. We will consider incorporating some non-convergence simulation results. Thank you for your suggestion.

---

### Official Review · Reviewer_NE2Q · 2022-07-15

**Rating:** 7
**Confidence:** 3
**Soundness:** 4 excellent
**Presentation:** 4 excellent
**Contribution:** 4 excellent

**Summary:**

This paper considers the problem of multi-task peer prediction where the goal is to elicit truthful information from agents using a scoring mechanism that scores agents based on their joint reports. It connects this problem of multi-task peer prediction to online no-regret learning by considering agents that learn to manipulate their reports in order to maximize their scores. The paper considers a special case of two agents and binary signal/reports and formulates the problem as a two-player repeated game. The paper shows that given any scoring mechanism there exist no-regret algorithms for agents which do not converge to truthful reporting. The paper then considers the CA mechanism which is a well-known mechanism for peer prediction and shows that a large class of learning algorithms converge to truthful reporting. This large class includes several no-regret algorithms such as Hedge, FPL etc. The paper also shows empirically that many popular online learning algorithms converge to truthful reporting.

**Questions:**

1. Do the results presented in this paper extend to the case of multiple agents? It is not clear how to define a learning scenario for multiple agents and define the CA mechanism under such a learning scenario.

2. It would be very interesting to see if the results in this paper can be extended to the case of non-binary signal/reports?

3. It would also be interesting to understand necessary conditions that result in truthful convergence of learning algorithms under the CA mechanism.

**Limitations:**

I do not foresee any negative societal impact of this paper.

**Strengths And Weaknesses:**

Strengths: I think that this paper provides a significant contribution to the burgeoning literature on mechanism design for learning agents. The problem of peer prediction is an important problem in information elicitation and is well-motivated due to various applications in crowdsourcing. The paper initiates a study of the dynamics of learning under this important problem and provides several non-trivial results.

Weakness: The paper considers a very simple setting with two agents and binary reports. Nevertheless, I think that this simple setting lays a solid foundation to build upon in the future.

---

> ### Author Response · Authors · 2022-08-02
> **Response to Reviewer NE2Q**
>
> Thank you very much for your time and effort in reviewing our paper. We appreciate your comments.
>
> **Extend to the case of multiple agents**
>
> There is a straightforward extension of our work to the setting of multiple agents. We can randomly pair the learning agents and run CA mechanisms on each pair. Our current results imply that every pair of agents using algorithms from the family $\mathcal{A}$ will achieve truthful convergence. Moreover, if the number of agents is odd, for the remaining agent, we randomly pick another agent as her reference agent and reward this remaining agent according to the CA mechanism as if she plays against her reference agent.  We added the above discussion as a footnote in our revised paper.
>
> **Extend to the case of non-binary signal/reports**
>
> During the simulation, we observed that when the signals are non binary (ternary), Alice and Bob using FPL may converge to nontruthful (and non-permutation) strategies in the CA mechanism.   This result indicates that using reward-based online learning algorithms is not sufficient for truthful convergence under the CA mechanism.
>
> Additionally, previous work by Shnayder et al. also shows similar nontruthful convergence holds when agents use continuous time learning dynamics when the signals are non-binary.
>
> We believe extending to nonbinary settings is an exciting future direction but better mechanisms or stronger assumptions than positively correlated assumptions are needed.
>
> **Necessary conditions that result in truthful convergence of learning algorithms under the CA mechanism**
>
> Thank you for this suggestion. We have incorporated a necessary condition in our revised paper. We have proved that truthful convergence under the CA mechanism implies that both agents are *no-regret in the game*. In other words, “both agents being *no-regret in the game*” is a necessary condition of truthful convergence. Please refer to theorem 3.2 in our revised version.
>
> On the other hand, both agents being no-regret is not a sufficient condition for truthful convergence as shown by theorem 3.1. Our work shows that using reward-based online learning algorithms is a sufficient condition for truthful convergence. A more systematic study on the gap between the necessary conditions and the sufficient conditions is an interesting future direction.

---

### Meta-Review · Area_Chair_TNAM · 2022-08-24

**Recommendation:** Accept
**Confidence:** Less certain

**Metareview:**

The reviews are mostly positive and consider that the paper solves an interesting problem with non-trivial techniques. For further improvement, the reviewers mentioned some concerns about the problem setup being restricted, and gave some suggestions for improving presentation.

**Award:**

No

---

### Decision · Program_Chairs · 2022-09-14

Accept